# Extracting and Following Paths for Robust Relational Reasoning with Large Language Models

**Ge Zhang**                                                    *ge.zhang1@huawei.com*
*Huawei Technologies Canada*

**Mohammad Ali Alomrani**                    *mohammad.ali.alomrani@huawei.com*
*Huawei Technologies Canada*

**Hongjian Gu**                                            *hongjian.gu@huawei.com*
*Huawei Technologies Canada*

**Jiaming Zhou**                                         *jiaming.zhou@huawei.com*
*Huawei Technologies Canada*

**Yaochen Hu**                                            *yaochen.hu@huawei.com*
*Huawei Technologies Canada*

**Bin Wang**                                              *wangbin158@huawei.com*
*Huawei Technologies*

**Qun Liu**                                                  *qun.liu@huawei.com*
*Huawei Technologies*

**Mark Coates**                                            *mark.coates@mcgill.ca*
*McGill University*
*Mila - Québec AI Institute*

**Yingxue Zhang**                                       *yingxue.zhang@huawei.com*
*Huawei Technologies Canada*

**Jianye Hao**                                            *jianye.hao@huawei.com*
*Huawei Technologies*

**Reviewed on OpenReview:** *https://openreview.net/forum?id=EbELaNKmZK*

## Abstract

Large language models (LLMs) possess vast semantic knowledge but often struggle with complex reasoning tasks, particularly in relational reasoning problems such as kinship or spatial reasoning. In this paper, we present Path-of-Thoughts (PoT), a novel framework for solving relation reasoning that decomposes the task into three key stages: graph extraction, path identification, and reasoning. Unlike previous approaches, PoT efficiently extracts a reasoning graph that identifies crucial entities, relations, and attributes within the context. Subsequently, PoT identifies query-relevant reasoning paths within the graph, facilitating downstream reasoning of potential answers. Experimental evaluations across four datasets of relational reasoning demonstrate that PoT surpasses state-of-the-art baselines by a significant margin (up to 21.3%) without requiring fine-tuning or extensive LLM calls. Furthermore, unlike prior neuro-symbolic methods, PoT exhibits improved resilience against LLM extraction errors and input ambiguity by leveraging the compositional nature of graphs.

# 1 Introduction

Large language models (LLMs) have shown remarkable generalization abilities in natural language (NL) tasks (Wei et al., 2022a; Kojima et al., 2022), such as generating useful code (Chen et al., 2021) and fluently engaging in dialogue (Thoppilan et al., 2022). Their success can be attributed to pre-training on large human language datasets, which express real-world concepts, and thereby allow LLMs to implicitly learn about the entities and relations that exist in the physical world (Patel & Pavlick, 2022). Nonetheless, some argue that the underlying meaning of language cannot be learned from text alone without appropriate grounding to the (non-text) real-world experiences (Bisk et al., 2020; Cohn & Blackwell, 2024). Prior studies (Tolman, 1948; Whittington et al., 2022; Garvert et al., 2017) have shown that humans can create "cognitive maps" while navigating and experiencing their environments. Cognitive maps represent the latent relational structure of a task/environment and are particularly helpful for multi-hop relational reasoning tasks such as planning/navigation (Yamada et al., 2024; Momennejad et al., 2023).

While LLMs do exhibit some competence in basic planning tasks (Momennejad et al., 2023; Valmeekam et al., 2023), they are known to perform shallow reasoning and suffer in multi-hop relational reasoning tasks (e.g., kinship inference (Sinha et al., 2019), or spatial reasoning (Shi et al., 2022)). In contrast, symbolic solvers (e.g., Answer Set Programs (ASP) (Lifschitz, 2008)) can faithfully perform reasoning using well-defined symbolic rules written by domain experts. Consequently, there has been a surge of neuro-symbolic works (Yang et al., 2023; Mirzaee & Kordjamshidi, 2023; Silver et al., 2024; Pan et al., 2023) which combine the rich LLM natural language abilities with interpretable symbolic solver modules. These works typically leverage LLMs to transfer any natural language (NL) based problem formulation to the appropriate symbolic language. This is then executed by the solver, hence maintaining the flexibility of LLMs while transferring the burden of complex reasoning to the symbolic reasoning module. This disentanglement of language understanding and reasoning displays significant performance improvements over prompt-based baselines (e.g., Chain-of-thought (CoT) (Wei et al., 2022b)). This idea of task decomposition also demonstrates the benefits in other research domains (Wang et al., 2023a). Nonetheless, prior works suffer from several shortcomings, such as task-specific and highly specialized translation and reasoning modules, brittleness to LLM errors, or requiring many LLM calls.

In this work, we introduce a novel framework, Path-of-Thoughts (PoT), that decomposes a relational reasoning problem into three stages: graph extraction, path identification, and reasoning. During the first stage, a single LLM call extracts the key entities, relations, and their corresponding attributes in the problem, constructing a graph (akin to a cognitive map), which serves as a foundation for downstream relational reasoning tasks (e.g., kinship or spatial). Subsequently, the path identification module identifies the key reasoning paths in the graph that are associated with the question. Finally, an LLM or symbolic reasoner is used to infer probable answers based on the input and identified paths. Our evaluations across several well-established relational reasoning datasets indicate a maximum improvement of 21.3% in accuracy and superior robustness to LLM extraction errors and potential controversial descriptions. *To the best of our knowledge, PoT is the first work that enhances relational reasoning tasks through identifying query-relevant subgraphs before reasoning.* Our contributions can be summarized as follows:

- We present a prompting-based approach to efficiently extract graphs and queries in a single LLM call.

- We propose a path identification stage that can identify multiple independent reasoning paths involving the queried entities to infer all possible answers.

- We benchmark on several kinship and spatial reasoning datasets to validate the performance gain of proposed framework against baselines.

# 2 Problem Definition

In relational reasoning, a sample $(s, a)$ consists of a textual story $s$ and a target relation $a \in \mathcal{R}$, where $\mathcal{R}$ is the overall set of pre-defined relations. A story $s$ consists of a context $c$ and a question $q$. The context $c$

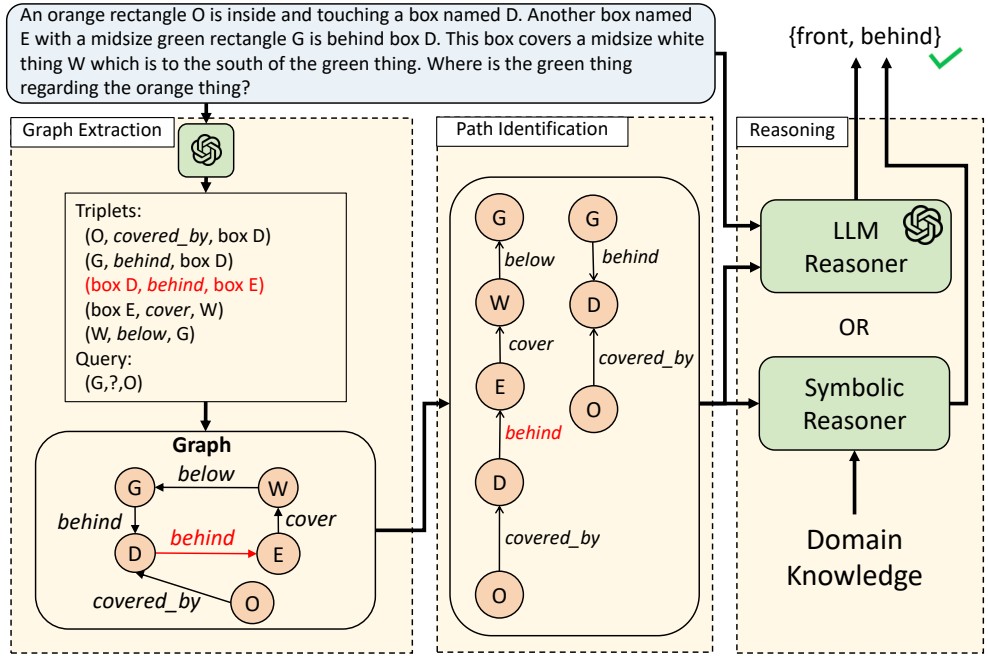

Figure 1: The PoT framework with an example featuring robustness against LLM extraction errors highlighted in red (flipped relation). The LLM is employed to extract the graph representing the story's relational structure. Path identification isolates the reasoning paths relevant to the query entities. PoT reasons over each path independently to alleviate cascading errors due to extraction and infers all possible answers.

describes relations among a set of entities $\mathcal{N}$ in natural language (e.g., A is the son of B), where the relation set mentioned in context $c$ is a subset of $\mathcal{R}$. The question $q$ asks about the relation between 2 entities, $n_i, n_j \in \mathcal{N}$ in natural language (e.g., How should B address C?), whose relation is not directly provided in the context $c$. Note that many prior works (Yang et al., 2023; Mirzaee & Kordjamshidi, 2023) assume story $s$ only consists of context $c$ without question $q$, and 2 queried entities $n_i$ and $n_j$ are provided by the dataset instead of being extracted from the question $q$. We target a more general setting in which the dataset does not annotate 2 queried entities and must extract them from the question $q$ for the story $s$.

For some datasets, there could be multiple possible answers due to ambiguities or errors in the story (See Appendix A.8 for examples). Therefore, a method is allowed to output multiple possible relations.

To complete the task, the algorithm must understand how relations combine (e.g., if A is to the west of B and C is north of A, then C is north-west of B). In the problems we address, we assume that these compositions are either common-sense (and thus implicitly encoded in an LLM) or that a domain-specific rule set is provided, specified as logical rules or a set of examples.

## 3 Related Work

**Multi-hop Relational Reasoning:** Before the advent of LLMs, several neural network architectures were proposed to solve the relational reasoning problem. The introduction of benchmark datasets often accompanies these. Shi et al. (2022) introduces the StepGame dataset, which tests for multi-hop spatial reasoning. That is, given a story describing the spatial relations (on top of, down, right, etc.) between entities, the task is to infer the implicit relation between two entities in the story. The authors introduce the Tensor-Product based Memory-Augmented Neural Network (TP-MANN), which is based on memory networks (Schlag et al., 2021) and specialized for spatial reasoning tasks. Palm et al. (2018) design a relational recurrent network, which treats the input relational problem as a fully connected graph with nodes representing the facts. Message passing is iteratively performed before the answer is predicted. Recent

methods (Mirzaee et al., 2021; Mirzaee & Kordjamshidi, 2022; 2023) fine-tune pre-trained language models (PLMs) (e.g., BERT (Devlin et al., 2019)) to extract richer textual features and cast the problem into a sequence classification task. Wang et al. (2023b) provides another synthesized dataset called SPARTUN for testing spatial reasoning problems. Compared to the StepGame dataset, it includes a larger variety of spatial relation types and expressions. The authors fine-tune a PLM-based model with a classification layer on top of it to predict the final relation between two queried entities. Sinha et al. (2019) introduces the CLUTRR dataset to benchmark the kinship reasoning abilities of NLP models. Experimental results show that a large gap exists between PLMs that reason directly on the textual input and graph neural network models (Veličković et al., 2018) that work directly on the underlying symbolic graph manifested by the story.

**Prompting-based Reasoning Methods:** With the emergence of powerful LLMs like GPT-4 (Achiam et al., 2024) and GPT-4o, many approaches leverage the natural language understanding and reasoning capabilities of these models to tackle multi-hop relational reasoning problems. Wei et al. (2022b) introduces Chain-of-Thought (CoT) prompting, which instructs LLMs to reason step-by-step before arriving at conclusions. A follow-up work, Chain-of-Thought with Self Consistency (CoT-SC) (Wang et al., 2023b), seeks to improve CoT by performing multiple independent reasoning iterations followed by a majority vote. Subsequent frameworks, such as Tree-of-Thoughts (ToT) (Yao et al., 2024) and Graph-of-Thoughts (GoT) (Besta et al., 2024), further enhance LLMs' reasoning capabilities on specific downstream tasks by utilizing sophisticated search strategies and task-specific heuristics (e.g., scoring functions). However, these tailored search strategies limit their adaptability to other downstream tasks (e.g., relational reasoning). To address these challenges, Zhou et al. (2024) proposed using graph-based synthetic data to fine-tune LLMs, along with an Extract-then-Answer prompting strategy. This approach showed improved performance in inductive and spatial reasoning tasks by leveraging structured reasoning representations. Additionally, Hu et al. (2024) proposed Chain-of-Symbol (CoS) prompting to address spatial reasoning problems by presenting LLMs with in-context examples that include stories and corresponding symbolic chains. However, CoS still relies on LLMs to not only translate natural language into symbolic notations but also to implicitly construct reasoning paths, which makes it vulnerable to interfering or disordered relations within the input relational reasoning problem.

**Extraction and Symbolic Reasoning:** The interpretability (Singh et al., 2024) and hallucination (Huang et al., 2025) issues of LLMs have led many works to complement them with symbolic modules (Pan et al., 2023; Olausson et al., 2023; Nye et al., 2021; Wong et al., 2023b;a; Yu et al., 2023). Such neuro-symbolic systems have been successfully applied to visual question answering (Ding et al., 2021) and robot planning (Wong et al., 2023b; Silver et al., 2024; Yang et al., 2023). DSR-LM (Zhang et al., 2023) presents a differentiable symbolic reasoning framework that uses pre-trained language models for fact extraction alongside a differentiable symbolic module for deductive reasoning using learned rules. The method displays good performance on kinship reasoning but requires significant finetuning and can fail due to fact extraction errors. LINC (Olausson et al., 2023) introduces a framework for first-order logic (FOL) reasoning that employs LLMs as semantic parsers to translate natural language premises and conclusions into first-order logic expressions. Subsequently, external theorem provers are used for deductive inference. This approach leads to significant performance improvements over pure prompting-based methods. However, it is limited to first-order logic problems that are expressed in relatively short statements, which makes the semantic parsing task tractable. Logic-LM (Pan et al., 2023) also employs LLMs for semantic parsing but tackles more logic-oriented problems, such as logic programming and constraint satisfaction. LLM-ASP (Yang et al., 2023) uses answer set programs (Lifschitz, 2008) as generic symbolic solvers, resulting in a versatile system capable of achieving state-of-the-art performance across various problems. Our framework, PoT, embraces the neuro-symbolic paradigm, but has important, distinct features. Rather than converting the input problem into task-specific symbolic language, we opt to extract the fundamental entities and relations, constructing a versatile graph that can be utilized by a variety of downstream reasoners (e.g., LLM, symbolic solver, etc). Unlike other complex symbolic formats, graphs offer support for *compositional* and easily *interpretable* reasoning, making them particularly suitable for tasks rooted in relationships, such as spatial reasoning. Moreover, we refrain from imposing any specific format assumptions on the input text problem. Rather, we efficiently extract all relations and queries simultaneously within a single LLM call. Lastly, while traditional symbolic solvers may fail if given contradictory facts, the inherent compositional nature of graphs in PoT enables us to mitigate the impact of conflicting information due to LLM extraction errors or ambiguities in the input problem.

By exploring *multiple reasoning paths* between queried entities, our approach offers resilience against such challenges (See Figure 1 for an example). To the best of our knowledge, our approach is the first that directly mitigates the effect of LLM extraction errors on the reasoning module.

## 4 Methodology

The proposed framework, PoT, consists of 3 modules: graph extraction, path identification, and reasoning. The graph extraction module extracts all mentioned entities and relations with corresponding attributes from the input story with LLMs, and later converts them into a graph. Subsequently, the path identification module identifies all reasoning paths between the two queried entities on the graph. Lastly, the reasoning module infers the answer given for each reasoning path independently.

Figure 1 shows the overall diagram of the proposed framework. Section 4.1 elaborates on how to prompt LLMs to extract the graph effectively. Section 4.2 describes the process of finding the relevant reasoning paths between the queried entities on the graph. Section 4.3 introduces how we employ either an LLM or a symbolic solver to infer the final answers given the reasoning paths.

### 4.1 Graph Extraction

Given a textual input story $s$, the objective of the graph extraction module is to convert the context of the story $s$ into a graph $\mathcal{G} = (\mathcal{N}, \mathcal{E})$, where the node set $\mathcal{N} = \{n_1, n_2, \ldots, n_k, \ldots\}$ represents entities in the context, with their associated attributes, and the edge set $\mathcal{E} = \{e_1, e_2, \ldots, e_k, \ldots\}$ contains triplets represented as $(n_{head}, r, n_{tail})$, where head node $n_{head}$ and tail node $n_{tail}$ represent entities, and $r$ denotes the relation from $n_{head}$ to $n_{tail}$. Note the relation $r$ belongs to the pre-defined relation set $\mathcal{R}$. For example, in the kinship domain, a node includes attributes like 'identity' and 'gender', representing a person's name and gender, respectively. This section details our approach to constructing effective prompts for graph extraction using large language models (LLMs).

Despite the effort of few-shot prompting (Brown et al., 2020), a significant challenge in graph extraction lies in the potential for the LLM to misinterpret the textual input, leading to missing nodes or incorrect relations. These inaccuracies can compromise the reliability of the graph $\mathcal{G}$, ultimately affecting the reasoning tasks that depend on it.

To address these challenges, we designed prompts that explicitly guide the LLM toward accurate relation identification and triplet extraction. Our approach builds on principles of structured guidance and decomposition, adapting strategies from prior works while introducing specific enhancements tailored to the graph extraction task. Key components of our methodology include: (i) *Sectional markup for logical structure*, (ii) *Syntactic delimiters for output consistency*, (iii) *Predefined categories for standardized outputs*, and (iv) *A decomposed approach to task simplification*. Examples of the prompts developed for our experiments are detailed in the Appendix A.5.

**Structured Prompts with Sectional Markup.** Inspired by previous work (Zhong et al., 2022) that organizes prompts into logically segmented sections to improve interpretability, we structure our prompts with distinct sections marked by special characters (for example, #). This logical organization provides the LLMs with a clear and navigable framework, reducing ambiguity during task processing.

**Structured Output with Syntactic Delimiters.** Inspired by the method proposed by Zhong et al. (2022) that uses logical segmentation for clarity, we systematically organize prompts into distinct sections, marked with special characters (e.g., #). This structure allows the LLM to navigate the task more effectively and minimizes ambiguity in interpreting the input.

**Predefined Output Categories.** Following principles of consistent formatting, we use syntactic markers such as brackets or parentheses to enforce a standardized output format. This approach ensures precision in the extracted data, reducing the likelihood of parsing errors during downstream processing.

**Decomposition of the Extraction Task.** Similar to the prompt ideas outlined by Li et al. (2023) and Wu et al. (2022), we decompose the graph extraction process into smaller subtasks. For example, the prompt separates the generation of relational triplets from the queries identifying the two nodes. This explicit task decomposition reduces the cognitive load on the LLM, enabling it to focus on individual subtasks and improving overall performance.

By integrating these strategies, we tailored the LLM prompts to balance clarity, consistency, and task-specific adaptability, enabling effective graph extraction across diverse domains.

The output of the LLM is parsed into in a set of triplets $\mathcal{E}$ which is the edge set of the graph $\mathcal{G}$. The queried entities from the question are also extracted as nodes on graph $\mathcal{G}$, represented as $n_{src}$ and $n_{tar}$, respectively.

## 4.2 Path Identification

The path identification module is responsible for identifying all reasoning paths on the graph $\mathcal{G}$. A reasoning path $\boldsymbol{p}$ is a sequence of edges on $\mathcal{G}$ that connects the query nodes $n_{src}$ and $n_{tar}$. Specifically, $\boldsymbol{p} = [e_i, \ldots, e_j]$, where, $e_i, e_j \in \mathcal{E}$. Note that the direction of the edge $e_i$ between any adjacent nodes $n_s, n_k \in \mathcal{N}$ can be either forward as $e_i = (n_s, r, n_k)$ or backward as $e_i = (n_k, r, n_s)$, depending on which of them exists in edge set $\mathcal{E}$. We apply depth-first search (Sedgewick, 2001) to identify such reasoning paths on the graph $\mathcal{G}$ between the given queried entities.

Intuitively, a single reasoning path is all that is needed to infer the implicit relation between $n_{src}$ and $n_{tar}$ (See Figure 1 for example). However, there could be multiple possible reasoning paths from $n_{src}$ to $n_{tar}$. In cases where there are LLM extraction errors or ambiguities in the story, each reasoning path can infer a different possible answer (i.e., relation).

## 4.3 Reasoning

For each reasoning path $\boldsymbol{p}$, we call an external reasoner (e.g., symbolic solver) to obtain the target relation $a \in \mathcal{R}$. The choice of reasoner depends on whether domain-specific rules (e.g., logic rules) are available and other user considerations (e.g., speed, robustness, optimality). In this work, we explore both LLM and symbolic reasoners.

**LLM Reasoner:** The LLM directly infers the answer given the input problem, query, and extracted reasoning path (expressed in natural language). This assumes that the LLM has common-sense knowledge of the problem at hand (e.g., spatial rules). Unlike Chain-of-Thought prompting (Wei et al., 2022b), which asks the LLM to perform step-by-step reasoning before answering, we explicitly extract the reasoning path relevant to the query in the path identification stage, before feeding it to the LLM for reasoning. This alleviates common issues suffered by LLMs due to irrelevant context (Shi et al., 2023).

**Symbolic Reasoner:** We use the CLINGO solver (Lifschitz, 2019) which is based on answer set programming (ASP) (Lifschitz, 2008). ASP is a logic programming paradigm that is effective for various knowledge-intensive reasoning tasks, particularly difficult (NP-Hard) search problems. Using CLINGO requires defining ASP knowledge modules which outline the rules needed to solve the problem at hand (e.g., `grandson(a,b)` $\wedge$ `sister(b,c)` $\implies$ `granddaughter(a,c)`). Each edge in the extracted reasoning path is translated to a fact represented in ASP language (e.g., (John, brother, Jack) $\rightarrow$ `brother(Jack, John)`) The solver infers the answer given the facts and rules (i.e., problem-specific knowledge module). See Appendix A.9 for details.

# 5 Experiments

## 5.1 Experimental Setup

We evaluate all methods on four relational reasoning datasets: *(i)* **StepGame** (Shi et al., 2022): A QA benchmark aiming to evaluate spatial reasoning abilities. This dataset contains a controllable parameter $k$ which specifies the possible length of reasoning hops. We use 1000 samples for each $k \in \{3, 4, 10\}$. *(ii)*

**CLUTTR** (Sinha et al., 2019): A banchmark for evaluating the English kinship reasoning abilities. We used the test set provided by the author at huggingface [1]. Within this test set, the number of reasoning hops required to infer the answer ranges from 2 to 10. The final test dataset has 1049 samples. Each sample consists of the context, query, and label. The context describes relationships among persons within a family in a natural tone. The query provides the names of the two persons whose relation we need to deduce. The label contains the answers to the query. Moreover, the context has names tagged within '[]' and the queried entities are known and not part of the story, which is inconsistent with our problem definition (See Section 2). Therefore, we modified the test set by canceling name tagging and merging the query into the story as a natural language question (e.g., How should x address y?). *(iii)* **SPARTUN** (Mirzaee & Kordjamshidi, 2022): A synthesized dataset created for spatial question answering. It has a broad coverage of various types of spatial relations, spatial language expressions, and utterances. It has 2 types of questions: *Find Relation* and *YES/NO*. We run the experiments on the *Find Relation* type of questions as they return the actual relationships, while *YES/NO* questions do not. We use the first 1000 *Find Relation* questions from the test set in all experiments. The original dataset has its label in one of 15 concepts: 'FAR', 'NTPP', 'EC', 'NTPPI', 'TPP', etc. To make it easier for LLMs to understand the label of the questions, we map the concepts back to their meaning in plain text. *(iv)* **Chinese Kinship**: We employed annotators to manually compile 73 story-and-answer pairs specifically focused on evaluating LLMs' ability for Chinese kinship reasoning. Chinese kinship is known for its complex relationships and presents a significant challenge for LLMs in conducting relational reasoning. Compared to English, Chinese kinship reasoning is more challenging for 3 reasons: 1) the ages of people affect the final kinship title; 2) there are over 500 possible titles; and 3) there are aliases for kinship titles due to regional customs.

**Baseline Methods:** We benchmark our method against a range of prompting-based and neuro-symbolic methods. We use standard Input-Output prompting (`Zero-shot`), Few Shot prompting (Brown et al., 2020), Chain-of-Thought (`CoT`) prompting (Wei et al., 2022b), and CoT with self consistency (`CoT-SC`) (Wang et al., 2023b) as prompting baselines. `Zero-shot` prompts the LLM to generate the answer directly, given an instruction and the input story. Few-shot prompting provides a few question-answer pairs as examples. `CoT` encourages LLMs to outline detailed reasoning steps before outputting the answer. Finally, `CoT-SC` repeatedly calls the LLM with the same prompt and outputs the most frequent answer. Both `CoT` and `CoT-SC` are with few-shot examples. To represent neuro-symbolic methods, we benchmark `LLM-ASP` (Yang et al., 2023), which first extracts symbolic facts from the story using LLMs and then uses ASP (Lifschitz, 2008) solvers for inferring answers. We choose `LLM-ASP` since it displays good performance on a variety of relational reasoning tasks and requires no finetuning. More details on `LLM-ASP` experiments can be found in Appendix A.9. We do not benchmark neuro-symbolic methods (e.g., `LLM-ASP`, PoT w/ symbolic reasoner (`PoT-Symbolic`)) on the Chinese kinship and SPARTUN datasets as the complexity of their possible relations (e.g., >500 possible Chinese kinship relations) makes it difficult to write a symbolic knowledge module (See Appendix A.11 for details).

**Backbone LLMs:** We benchmark all methods using GPT-3.5-turbo (0125) (Ouyang et al., 2022), GPT-4-turbo (2024-04-09) (Achiam et al., 2024), GPT-4o (2024-05-13) and Llama3-70B-instruct (Grattafiori et al., 2024). Besides the general-purpose LLMs, we also benchmark all methods on the CLUTTR (Sinha et al., 2019) and StepGame (Shi et al., 2022) dataset using reasoning-focused LLMs, namely GPT-o1-mini (Jaech et al., 2024), Claude-3.7-sonnet [2], and GPT-5 [3]. All experiments were conducted with a sampling temperature of 0.3 and a maximum output token length of 4096, except for GPT-5, where we set the sampling temperature and output token length to their default values.

**Performance Metrics:** We follow prior works to measure accuracy between predicted relations and ground truth relations, where it checks whether at least one target relation exists within the predicted relations.

---

[1]https://huggingface.co/datasets/CLUTRR/v1/viewer/gen_train234_test2to10/test
[2]https://www.anthropic.com/news/claude-3-7-sonnet
[3]https://openai.com/index/introducing-gpt-5/

## 5.2 Experimental Results

Table 1: Single-trial accuracy results. Prompting-based methods use LLMs to directly predict the answer. Neuro-symbolic methods (extraction + symbolic reasoning) use LLMs for semantic parsing and symbolic solvers for reasoning. `PoT-LLM` and `PoT-Symbolic` represent using an LLM or a symbolic solver as the reasoner, respectively. The $k$ for StepGame represents the number of reasoning hops required to infer the answer. The **bold** and underline fonts represent the best and second-best results, respectively. * means the **highest accuracy** is statistical significantly higher than the second highest accuracy measured by the paired McNemar test.

| LLM | Method | Stepgame k=3 | Stepgame k=4 | Stepgame k=10 | CLUTRR | SPARTUN | Chinese Kinship |
|---|---|---|---|---|---|---|---|
| GPT-3.5-turbo | *Prompting-based* | | | | | | |
| | Zero-shot | 24.0 | 22.5 | 17.0 | 31.2 | 44.3 | 20.5 |
| | Few-shot | 21.3 | 20.8 | 16.8 | 33.4 | 35.1 | 23.3 |
| | CoT | 31.1 | 26.7 | 19.3 | 35.6 | 44.2 | 21.9 |
| | CoT-SC | 30.7 | 28.0 | 21.7 | **37.1** | 47.6 | 24.7 |
| | PoT-LLM (Ours) | **50.9**\* | **44.8**\* | **28.8**\* | 35.1 | **52.7**\* | **27.4** |
| | *Extraction + Symbolic Reasoing* | | | | | | |
| | LLM-ASP | **76.4**\* | **83.7**\* | **72.6**\* | 32.8 | – | – |
| | PoT-Symbolic (Ours) | 72.4 | 75.9 | 66.0 | **54.1**\* | – | – |
| GPT-4-turbo | *Prompting-based* | | | | | | |
| | Zero-shot | 59.0 | 52.5 | 32.8 | 45.9 | 72.8 | 45.2 |
| | Few-shot | 55.3 | 50.7 | 29.8 | 42.0 | 76.9 | 37.0 |
| | CoT | 58.3 | 51.3 | 34.2 | 53.0 | **79.7** | 39.7 |
| | CoT-SC | 57.4 | 51.7 | 34.4 | 54.6 | 78.1 | 46.6 |
| | PoT-LLM (Ours) | **67.4**\* | **59.8**\* | **40.1**\* | **57.6**\* | 75.5 | **53.4** |
| | *Extraction + Symbolic Reasoing* | | | | | | |
| | LLM-ASP | 83.7 | 89.4 | 81.1 | 48.1 | – | – |
| | PoT-Symbolic (Ours) | **88.2**\* | **92.6**\* | **85.6**\* | **66.1**\* | – | – |
| GPT-4o | *Prompting-based* | | | | | | |
| | Zero-shot | 68.6 | 60.1 | 37.7 | 45.5 | 81.6 | 67.1 |
| | Few-shot | 36.6 | 36.4 | 26.6 | 36.5 | 80.5 | 65.8 |
| | CoT | 69.4 | 61.0 | 40.0 | 57.6 | 81.4 | 68.5 |
| | CoT-SC | 70.0 | 63.2 | 40.4 | 59.4 | 78.9 | 68.5 |
| | PoT-LLM (Ours) | **73.4**\* | **68.0**\* | **48.7**\* | **61.9**\* | **83.1** | **71.2** |
| | *Extraction + Symbolic Reasoing* | | | | | | |
| | LLM-ASP | 85.3 | 84.7 | 71.6 | 56.7 | – | – |
| | PoT-Symbolic (Ours) | **88.2**\* | **92.9**\* | **86.3**\* | **67.7**\* | – | – |
| Llama3-70B | *Prompting-based* | | | | | | |
| | Zero-shot | 24.4 | 20.0 | 10.1 | 24.2 | 65.0 | 31.5 |
| | Few-shot | 41.2 | 38.2 | 25.7 | 18.6 | 70.0 | **35.6** |
| | CoT | 60.0 | 50.6 | 31.6 | 52.8 | **73.8** | 27.4 |
| | CoT-SC | 61.3 | 50.2 | 32.9 | 54.7 | 72.6 | 27.4 |
| | PoT-LLM (Ours) | **69.1**\* | **61.6**\* | **41.8**\* | **56.3** | 72.8 | **35.6** |
| | *Extraction + Symbolic Reasoing* | | | | | | |
| | LLM-ASP | 85.7 | 89.6 | 83.2 | 54.8 | - | - |
| | PoT-Symbolic (Ours) | **86.4** | **92.6**\* | **84.5** | **65.9**\* | - | - |

**Full Pipeline Performance:** In Table 1, we compare the full pipeline performance of all baselines using different general-purpose backbone LLMs. For computational cost reasons, all results are single-trial. `PoT-LLM` and `PoT-Symbolic` represent using an LLM or a symbolic solver as the reasoner, respectively. For fair comparison, we compare `PoT-LLM` to prompting-based pipelines. On the other hand, `PoT-Symbolic` is compared to the `LLM-ASP` as extra domain knowledge (i.e., symbolic rules) is required for reasoning in both methods. The prompts we use can be found in Appendix A.3.

The results show a clear improvement of PoT over the baselines. Among the prompting-based methods, `PoT-LLM` outperforms all baselines in most scenarios, except the CLUTRR on GPT-3.5-turbo, and the SPAR-TUN on GPT-4-turbo and Llama3-70B. Meanwhile, for the extraction plus symbolic reasoning methods, results show a clear improvement of `PoT-Symbolic` over the `LLM-ASP`, with exceptions of the StepGame on GPT-3.5-turbo.

Interestingly, `CoT` and `Few-shot` prompting have only a minor improvement with powerful models (e.g., GPT-4) compared to `Zero-shot` prompting, as observed previously (Yang et al., 2023). This suggests that linear chain of thought reasoning may already exist in larger models, and imposing it externally is not always helpful for complex reasoning tasks. Moreover, we observe that the performance of most methods steadily degrades as the number of possible relations increases. Consequently, the Stepgame dataset shows the largest gap with prompting baselines, because directly solving this complex and high-order reasoning problem is too challenging for LLMs with just in-context learning examples.

More specifically, for GPT-3.5-turbo with prompting-based methods, `PoT-LLM` exceeds all baselines with a large margin except for the CLUTRR dataset. As for the extraction plus symbolic reasoning methods, the results show an improvement of `PoT-Symbolic` over the `LLM-ASP` on the CLUTRR dataset, but not on the Stepgame dataset. We believe the issue stems from the weaker instruction-following ability of GPT-3.5-turbo, which leads to poor triplet extraction outcomes. Better-optimized prompts could possibly improve performance.

For GPT-4o, employing prompting-based methods (`Zero-shot`, `Few-shot`, `CoT`, `CoT-SC`), shows significant improvement over GPT-4-turbo. This suggests an enhancement in its fundamental reasoning abilities, potentially due to training on a larger and more recent data corpus. Performance steadily decreases from $k = 3$ to $k = 10$ for Stepgame, except for neuro-symbolic methods, where $k = 4$ has the highest performance. We also observe that the improvements of our methods compared to the second-best methods increase when the number of reasoning hops increases. This observation holds for both prompting-based and extraction + symbolic reasoning, indicating that our method, with access to a powerful LLM, can outperform harder questions that require reasoning over longer reasoning chains.

As for Llama3-70b-instruct, our methods outperform or equal baselines in the majority of scenarios except for prompting-based on the SPARTAN dataset.

Moreover, we further evaluate our approach and the baseline methods on the CLUTRR and Stepgame dataset using reasoning-focused LLMs, GPT-o1-mini, Claude-3.7-sonnet, and GPT-5. The results are presented in Table 2. For GPT-o1-mini and Claude-3.7-sonnet, results show that our approach outperforms the baselines in both prompting-based and extraction-plus-symbolic settings, with a large margin in most scenarios, demonstrating the compatibility of the proposed method with these 2 reasoning LLMs. As for GPT-5, we notice that its reasoning effort is adjustable, which could affect the reasoning task we are trying to solve. Therefore, we try GPT-5 under two reasoning efforts: minimal and medium. Although our method performs on par with baselines under medium reasoning effort, it shows improvements when the reasoning effort is minimal, requiring much fewer tokens overhead than in medium reasoning effort. This demonstrates that our design effectively preserves reasoning accuracy even with reduced model reasoning effort.

**Computational Cost:** To evaluate the cost of experiments in detail, we record the average times and tokens spent on each module over all samples from the k=3 subset of the Stepgame dataset. The results are shown in Table 3 and Table 4, respectively.

Despite requiring two LLM calls, `PoT-LLM` achieves computational cost and latency comparable to IO and CoT. We attribute this efficiency to the simplicity of our extraction prompt and the modular design of

Table 2: Results on the Stepgame and CLUTRR datasets with reasoning-focused LLMs. * means the **highest accuracy** is statistical significantly higher than the second highest accuracy measured by the paired McNemar test

| LLM | Method | Stepgame | | | CLUTRR |
|---|---|---|---|---|---|
| | | k=3 | k=4 | k=10 | |
| *Cluade-3.7-sonnet* | *Prompting-based* | | | | |
| | Zero-shot | 70.6 | 63.5 | 37.8 | 47.9 |
| | Few-shot | 71.4 | 64.1 | 40.9 | 44.2 |
| | CoT | 76.0 | 70.2 | 44.5 | 57.4 |
| | CoT-SC | 75.3 | 70.6 | 46.0 | 61.7 |
| | PoT-LLM (Ours) | 81.0* | 76.9* | 57.2* | 67.7* |
| | *Extraction + Symbolic Reasoing* | | | | |
| | LLM-ASP | 84.9 | 89.8 | 83.7 | 67.3 |
| | PoT-Symbolic (Ours) | 88.0* | 92.6* | 87.8* | 67.6 |
| *GPT-o1-mini* | *Prompting-based* | | | | |
| | Zero-shot | 67.5 | 63.5 | 42.0 | 58.4 |
| | Few-shot | 67.5 | 57.9 | 38.3 | 61.3 |
| | CoT | 68.5 | 61.8 | 37.5 | 61.0 |
| | CoT-SC | 69.3 | 59.8 | 39.8 | **64.6** |
| | PoT-LLM (Ours) | 72.9* | 65.8* | 49.3* | 58.5 |
| | *Extraction + Symbolic Reasoing* | | | | |
| | LLM-ASP | 81.1 | 87.0 | 63.5 | 20.2 |
| | PoT-Symbolic (Ours) | **82.8** | **88.3** | 80.3* | 65.9* |
| *GPT-5-Medium* | *Prompting-based* | | | | |
| | Zero-shot | 89.1 | 88.2 | 81.4 | **73.2** |
| | Few-shot | 89.2 | 89.3 | 82.2 | 67.4 |
| | CoT | 89.4 | 89.0 | 82.0 | 70.2 |
| | CoT-SC | 89.3 | **89.7** | 83.0 | 71.1 |
| | PoT-LLM (Ours) | **89.5** | 89.5 | **83.7** | 71.5 |
| | *Extraction + Symbolic Reasoing* | | | | |
| | LLM-ASP | **89.2** | **93.8** | **88.1** | 44.0 |
| | PoT-Symbolic (Ours) | **89.2** | 93.3 | 87.8 | 72.3* |
| *GPT-5-Minimal* | *Prompting-based* | | | | |
| | Zero-shot | 44.7 | 40.9 | 27.5 | 28.6 |
| | Few-shot | 45.1 | 40.1 | 29.6 | 31.9 |
| | CoT | 69.3 | 76.9 | 60.5 | 55.2 |
| | CoT-SC | 73.7 | 82.7* | 67.4 | 60.7 |
| | PoT-LLM (Ours) | **75.6** | 75.5 | **68.9** | 65.0* |
| | *Extraction + Symbolic Reasoing* | | | | |
| | LLM-ASP | 82.4 | 87.6 | 82.1 | 2.3 |
| | PoT-Symbolic (Ours) | 88.8* | 93.2* | 87.8* | 65.4* |

our pipeline. Specifically, the reasoning module can operate on the reasoning path identified by the path identification module, rather than processing the entire input story. This targeted approach significantly reduces computational overhead while preserving strong performance.

**Graph Extraction Performance:** To evaluate the impact of prompts on the performance of the relation extraction, we construct a synthetic test set consisting of stories of multiple sentences and their corresponding triplets as labels. To balance the trade-off between manual labeling and data quantity for accurate results, we

Table 3: Detailed statistics of average time, in seconds, spent on each module. The results are calculated on all samples from the k=3 subset of the Stepgame dataset with GPT-4o as backbone LLM. $< t$ means actual time is less than $t$.

| Method | Modules | | | Overall |
|---|---|---|---|---|
| | **Graph Extraction** | **Path Identification** | **Reasoning** | |
| Zero-shot | - | - | - | 2.925 |
| CoT | - | - | - | 3.029 |
| PoT-LLM (Ours) | 1.363 | $< 0.001$ | 2.049 | 3.412 |
| PoT-Symbolic (Ours) | 1.337 | $< 0.001$ | 0.727 | 2.064 |

Table 4: Detailed statistics of average LLM tokens spent on each module. The results are calculated over all samples from the k=3 subset of the Stepgame dataset with GPT-4o as backbone LLM.

| Method | Graph Extraction | | Path Identification | | Reasoning | | Overall | |
|---|---|---|---|---|---|---|---|---|
| | **In Token** | **Out Token** | **In Token** | **Out Token** | **In Token** | **Out Token** | **In Token** | **Out Token** |
| Zero-shot | - | - | - | - | - | - | 249.5 | 196.6 |
| CoT | - | - | - | - | - | - | 2251.5 | 235.2 |
| PoT-LLM (Ours) | 1359.8 | 56.4 | - | - | 1285.6 | 147.8 | 2645.4 | 204.2 |
| PoT-Symbolic (Ours) | 1359.8 | 55.8 | - | - | - | - | 1359.8 | 55.8 |

manually labeled a pool of 100 sentences from the Stepgame dataset (Shi et al., 2022) with their corresponding triplets as the sentence pool. Each test story consists of 20 sentences uniformly sampled from the sentence pool and a query sentence asking about the spatial relation between 2 mentioned entities. The results of testing different prompts using GPT4-turbo on 1,000 such stories are presented in Table 5. The results demonstrate that the prompt design strategy we employ, which is explicitly tailored to extract relations (as detailed in Section 4.1), can accurately extract triplets from unstructured text. The introduced method significantly outperforms non-customized, in-context learning-based methods, such as zero-shot and CoT.

**Ablation Study on Path Identification Module:** To investigate the effectiveness of the path identification module, we conducted an ablation study by removing the path identification (P.I.) from both the prompting-based and symbolic-based variants of our framework. As shown in Table 6, eliminating P.I. leads to a notable performance drop for the prompting-based `PoT-LLM` pipeline, particularly on the Stepgame dataset, where accuracy decreases from 73.4 to 69.2 (`k=3`), 68.0 to 61.7 (`k=4`), and 48.7 to 38.8 (`k=10`), and on CLUTRR (61.9 to 50.7). This confirms that P.I. effectively guides the LLM to locate the correct reasoning chain and mitigates distraction from irrelevant paths. In contrast, the symbolic reasoning variant (`PoT-Symbolic`) shows only marginal changes (less than 0.5%), indicating that once explicit logical structures are extracted, path identification becomes less critical on the Stepgame and the CLUTRR dataset.

However, errors may arise from LLM graph extractions or controversial descriptions in the story. To better investigate the effectiveness of the path identification module, we built a synthetic dataset to simulate situations in which extraction errors occur during graph extraction, including but not exclusive to LLM errors or controversial descriptions in the story.

Based on observations of common LLM extraction errors (see Table 25), we design 7 possible noise types. Details can be found in Appendix A.1.

We build synthetic datasets based on the graph structural information underlying the stories that come with the CLUTRR dataset (Sinha et al., 2019). Details can be found in Appendix A.7. For each noise type, we generate 100 noisy samples. The results are shown in Figure 2. `Symbolic reasoner w/ P.I.` beats or ties the `symbolic reasoner w/o P.I.` for all noise types. `Symbolic reasoner w/o P.I.` is particularly sensitive to noise type like "adding irrelevant edge" and "adding main edge". Note that in the synthetic dataset, it is possible to introduce conflicting information since relations are chosen randomly. It is observed that `symbolic reasoner w/o P.I.` can struggle to resolve contradictions and is perturbed even if the conflicts are irrelevant to answer the question.

We also evaluate how `symbolic reasoner w/ P.I.` and `symbolic reasoner w/o P.I.` fare as more noise elements are introduced. We observe that `symbolic reasoner w/ P.I.` remains robust under various

Table 5: Extraction performance of different prompt techniques with GPT-4-turbo. 'Acc. Triplets' and 'Acc. Query' represents the percentage of correctly extracted triplets and queries among all stories, respectively. 'Acc. All' denotes the percentage of stories where the triplets and query were correctly extracted. The **bold** and underline fonts represent the best and second-best results within the group, respectively.

| Prompt Method | Acc. Triplets | Acc. Query | Acc. All |
|---|---|---|---|
| Zero-shot | 74.1 | 99.3 | 74.0 |
| CoT Zero-shot | 70.3 | 93.8 | 70.3 |
| Few-shot | 87.4 | 99.8 | 87.4 |
| CoT Few-shot | 91.9 | 99.9 | 91.9 |
| ours | **95.9** | **100.0** | **95.9** |

Table 6: Ablation study over the path identification module. The performance drops for the `PoT-LLM` pipeline and remains for `PoT-Symbolic` pipeline over Stepgame and the CLUTRR dataset.

| LLM | Method | Stepgame | | | CLUTRR |
|---|---|---|---|---|---|
| | | k=3 | k=4 | k=10 | |
| | *Prompting-based* | | | | |
| | `PoT-LLM` | 73.4 | 68.0 | 48.7 | 61.9 |
| | `PoT-LLM w/o P.I.` | 69.2 | 61.7 | 38.8 | 50.7 |
| *GPT-4o* | *Extraction + Symbolic Reasoing* | | | | |
| | `PoT-Symbolic` | 88.2 | 92.9 | 86.3 | 67.7 |
| | `PoT-Symbolic w/o P.I.` | 88.2 | 93.1 | 86.6 | 67.2 |

amounts of noise interference. In contrast, the performance of the `symbolic reasoner w/o P.I.` solver declines significantly as the amount of noise elements increases.

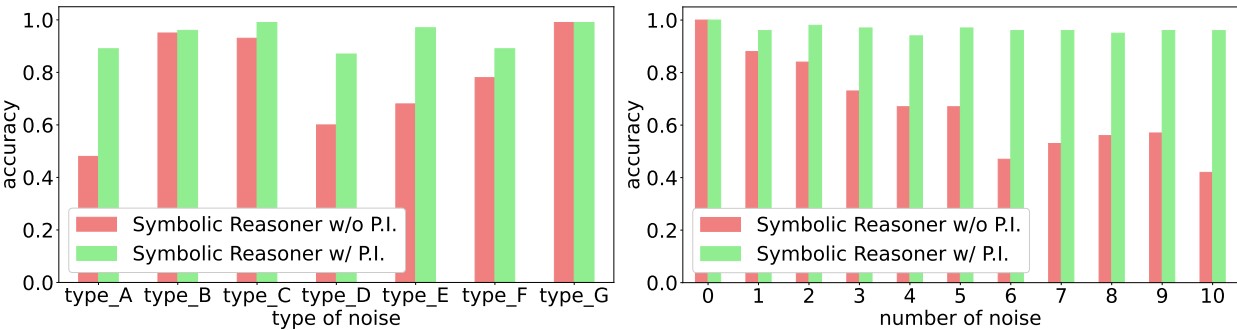

Figure 2: **Left:** Accuracy of `symbolic reasoner w/ P.I.` and `symbolic reasoner w/o P.I.` w.r.t noise Types. A: flip – irrelevant edge, B: add – new_node – one new edge, C: add – new_node – conflict edge, D: add – no_node – irrelevant edge, E: add – no_node – main edge, F: replace – irrelevant edge, and G: disconnected edges. **Right:** Accuracy of `symbolic reasoner w/ P.I.` and `symbolic reasoner w/o P.I.` w.r.t the number of noises.

## 6 Limitations

Although the proposed framework demonstrates promising performance, it has several important limitations.

First, PoT has been tested only on problems that satisfy the definition of relational reasoning, in which each relationship can be expressed as a composition or a chain of other relationships. The generality of PoT on non-relational reasoning tasks has not yet been systematically evaluated and remains an open question.

Second, the experimental setup is constrained by dataset quality. During our case studies, we identified multiple instances of incorrect or incomplete ground-truth labels that could confound evaluation results. Addressing these data-quality issues would likely further improve the reliability of our conclusions.

Third, constructing symbolic rules remains a challenge for `PoT-Symbolic`. This process requires domain expertise and significant manual effort. Recent work (e.g., (Lee & Hwang, 2025)) suggests that LLMs may help automate rule generation, an avenue we leave for future exploration.

Fourth, while `PoT-LLM` removes the need for manual rules, it introduces higher inference latency and a small accuracy drop due to its reliance on LLM-based reasoning. Advances in model architectures (e.g., MoE) and LLM training methods are likely to mitigate these issues, potentially enabling `PoT-LLM` to achieve both lower latency and stronger reasoning performance.

Given these complementary trade-offs between the PoT variants, we recommend `PoT-Symbolic` for domains with small, well-defined relations, where manual rules provide higher performance. In contrast, `PoT-LLM` is preferable in settings that require greater scalability and generality, despite its higher computational cost.

## 7 Conclusion

We introduce Path-of-Thoughts (PoT), a novel framework that decomposes a relational reasoning task into three stages: graph extraction, path identification, and reasoning. Our experiments demonstrate that PoT outperforms baselines across four benchmark datasets, without the need for fine-tuning or extensive large language model (LLM) calls. Unlike previous approaches, PoT exhibits strong resilience to noise relations by leveraging the compositional nature of graphs. Additionally, we conduct analysis experiments to demonstrate the contributions of each module of the PoT, and to highlight the importance of identifying key relations and the order of reasoning path in effective relational reasoning tasks.

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

# A Appendix

## A.1 Noise Types

We consider the graph to have two parts: a *main chain*, which is the primary reasoning path connecting the source node to the target node; and an *irrelevant part*, which consists of all nodes and edges that are not part of the main chain. When introducing noise, we do not corrupt the main chain, as we do not want to change the ground-truth answers. The 7 noise types are shown in Figure 3. Specifically, (A) **Flip an irrelevant edge**: Flip the direction of an irrelevant edge connecting 2 irrelevant nodes. (B) **Add a new node with a new edge**: Add a new node and a new edge that connects the new node to either the main chain or to a node in the irrelevant part of the graph. (C) **Add conflict edges**: Add a new node and connect it to either the main chain or the irrelevant part with 2 new edges. Noted that 2 new edges may contain conflict attributes. (D) **Add an irrelevant edge**: Add an edge connecting 2 irrelevant nodes. (E) **Add a main edge**: Add an edge between two nodes on the main chain. (F) **Modify the relation of an irrelevant edge**: Change the relation on an irrelevant edge without changing its direction. (G) **Add disconnected edge and nodes**: Add 2 new connected nodes that are both disconnected from the original graph.

## A.2 Path Counts Analysis

To better understand how multiple reasoning paths affect framework accuracy, we analyzed the distribution of path counts in the PoT-symbolic experiment on the CLUTRR dataset, using GPT-4o as the backbone LLM. The first two rows of Table 7 summarize both the path count distribution and the corresponding average accuracy. Most stories (697 out of 1048) produce only a single path, achieving an average accuracy of 68.9%. Accuracy increases as more paths are extracted, peaking at 84.6% when the path count reaches three, and then gradually declines as the path count continues to rise, reflecting the increasing difficulty of those samples.

We further conducted an experiment in which only the shortest path was identified and passed to the downstream symbolic solver. As shown in the last row of Table 7, this setting led to a notable drop in average accuracy across multiple path counts, reaching 14.3% drops for samples containing five or more paths. These results highlight the importance of capturing multiple reasoning paths in boosting final reasoning accuracy.

Table 7: Number of samples and average accuracy with respect to the number of chains extracted from the sample for the experiment of PoT-Symbolic over the CLUTRR dataset. The GPT-4o is applied as the backbone LLM.

| # of chains | 0 | 1 | 2 | 3 | 4 | 5+ |
|---|---|---|---|---|---|---|
| # of samples | 39 | 697 | 175 | 13 | 75 | 49 |
| Acc. of Identifying Multiple Paths (default) | 0.0 | 68.9 | 73.1 | 84.6 | 74.7 | 71.4 |
| Acc. of Identifying single Shortest Path | 0.0 | 68.9 | 63.4 | 84.6 | 66.7 | 57.1 |

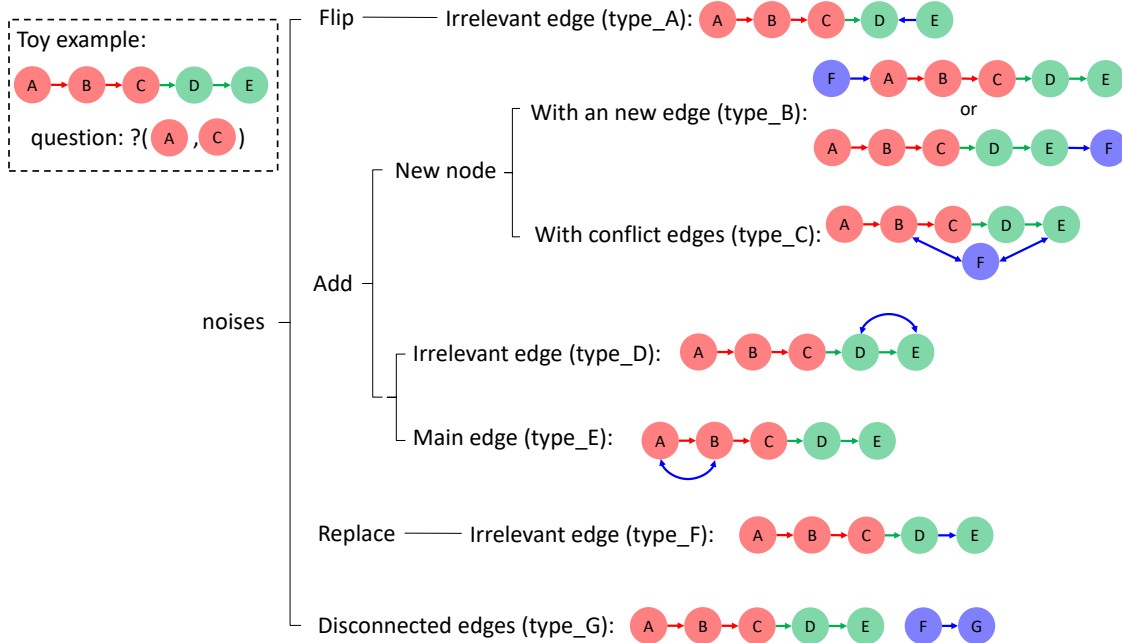

Figure 3: Illustrations of 7 noise types in the synthetic noise dataset. A toy graph with 5 nodes and 4 edges is shown. The nodes and edges that are relevant or irrelevant to answering the question are marked in red and green, respectively. The noisy nodes/edges are marked in purple.

### A.3 Implementation Details

**Compute**: All experiments were conducted using the OpenAI API[4] on an Intel(R) Xeon(R) Gold 6140 CPU @ 2.30GHz.

**ASP Solver** : For the neuro-symbolic methods, we use the Clingo ASP solver (Lifschitz, 2019). We utilize the ASP knowledge modules from LLM-ASP (Yang et al., 2023), which were written by human domain experts.

### A.4 Prompts for Baselines

In this section, we show the prompt templates we use for baselines in Table 1. The prompts are identical for each baseline across all backbone LLMs.

#### A.4.1 Prompt Templates for IO

The prompt templates of IO for StepGame, CLUTRR, SPARTUN, and Chinese kinship datasets can be found in Tables 8, 9, 10, and 11, respectively.

#### A.4.2 Prompt templates for Few-Shot

The prompt templates of Few-Shot for StepGame, CLUTRR, SPARTUN, and Chinese kinship datasets can be found in Tables 12, 13, 14, and 15, respectively.

---

[4]https://platform.openai.com/docs/introduction

Table 8: IO prompt template for StepGame.

| |
|---|
| Given a story about spatial relations among objects, answer the relation between two queried objects. |
| The answer could only be one of following: [top, bottom_left, top_left, bottom, bottom_right, top_right, right, left, overlap]. |
| If a sentence in the story is describing clock-wise information, then 12 denotes above, 1 and 2 denote upper-right, 3 denotes right, 4 and 5 denote lower-right, 6 denotes below, 7 and 8 denote lower-left, 9 denote left, 10 and 11 denote upper-left. If the sentence is describing cardinal directions, then north denotes above, east denotes right, south denotes below, and west denotes left. Wrap your final answer in brackets. Example: [top]. |
| |
| Story: {story} |
| Answer: |

Table 9: IO prompt template for CLUTRR.

| |
|---|
| Given a story about kinship relations among persons, answer the relation between two queried persons. The answer could only be one of following: [son, grandmother, daughter-in-law, grandson, greatgrandson, grandfather, mother-in-law, greatgranddaughter, uncle, son-in-law, wife, greatgrandfather, brother, husband, daughter, father-in-law, sister, greatgrandmother, granddaughter, aunt, nephew, niece, mother, father]. Wrap your final answer in brackets. Example: [grandfather] |
| |
| Story: {story} |
| Answer: |

### A.4.3 Prompt Templates for CoT and CoT-SC

The prompt templates of CoT and CoT-SC for StepGame, CLUTRR, SPARTUN, and Chinese kinship datasets can be found in Tables 16, 17, 18, 19, 20, and 21, respectively.

### A.4.4 Prompt Templates for LLM-ASP

The prompt templates of LLM-ASP for StepGame and CLUTRR (prompts for extracting relations and genders.) datasets can be found in Tables 22, 23, and 24.

### A.5 Prompt Templates for Graph extraction

Tables 27, 28, 29, and 30 showcase the prompt templates used for extracting the instance graph via in-context learning for the StepGame, CLUTRR, and Chinese kinship datasets, respectively.

**Common Extraction Errors:** We find GPT-4 models to be powerful information extractors. However, we do observe some common errors such as wrong gender, missing relations, or wrong entity. See Table 25 for examples from the CLUTRR dataset.

### A.6 Prompt Templates for LLM Reasoner

Tables 31, 33, 34, and 35 showcase the prompt templates used for LLM reasoner for the StepGame, CLUTRR, SPARTUN, and Chinese kinship datasets respectively. Note that for the Stepgame and CLUTRR datasets, we replace the original stories with the extracted instance paths, whose format can be found in the corresponding in-context learning examples. On the other hand, for the SPARTUN and Chinese kinship datasets, we append the extracted instance path together with the original story.

### A.7 Noise Datasets

We build synthetic noise datasets based on the clean samples in the CLUTRR dataset (Sinha et al., 2019), which we call the root sample set. When generating a noisy sample for a specific type of noise, we randomly select a sample from the root sample set and then add the noise to the sample at a random location. For each

Table 10: IO prompt template for SPARTUN.

Given a story about spatial relations among objects, answer the relation between two queried objects step by step. The answer could only be one of following: [far, in, touch, has, covered_by, right, overlap, front, behind, cover, left, disconnected_from, below, above, near]. 'inside and touching' refers 'covered_by'. 'inside' and 'within' and 'inside' refers 'in'. 'contain' refers 'has'. If the sentence is describing clock-wise information, then 3 denotes right, 6 denotes below, 9 denotes left, and 12 denotes above. If the sentence is describing cardinal directions, then north denotes above, east denotes right, south denotes below, and west denotes left. There could be multiple answers. Wrap all your answers in brackets. Example: [above, behind].

Story: {story}
Answer:

Table 11: IO prompt template for Chinese kinship.

You are given a question about chinese kinship relations, please answer the question step by step. ansaers include but not limited to chinese kinship titles: (从/表/堂)侄子,侄女,女婿,儿媳,岳父,岳母,妹夫,姐夫,伯公,叔公,etc. Wrap your final answer in square brackets []. If more than one relation is correct, separate the relations by comma, like: [舅表哥,舅表弟].

Question: {story}
Answer:

noise type, we generate 100 noisy samples. In all of these noise types, it is possible to introduce conflicting information (relations are chosen randomly).

To generate a sample containing $n$ noise elements, we first randomly select $n$ types of noise (with replacement), and then we introduce them into a randomly chosen sample from the root sample set. We generate 100 noisy samples for each number category. The findings are presented in Figure 2.

## A.8 CLUTRR Dataset Ambiguities

After manual checking, we have found several cases where the story has more than one possible answer. See Table 36 for examples in the CLUTRR dataset.

## A.9 Baselines

**LLM-ASP**: We use the same solver (CLINGO v5.6.0) and knowledge modules[5] used in LLM-ASP (Yang et al., 2023) for the CLUTRR and Stepgame datasets. Contrary to our problem definition (see Section 2), LLM-ASP (Yang et al., 2023) assumes that the query is given and need not be extracted for the CLUTRR dataset. Moreover, facts are extracted one sentence at a time for the stepgame dataset. Therefore, we modify the prompt so that the LLM extracts all triplets and queries with one LLM call.

**CoT-SC**: We use the same prompt as CoT for CoT-SC. We call LLMs 5 times and take the top 1 result of a majority vote as the final output of CoT-SC.

## A.10 More Experiments

## A.11 Chinese Kinship Complexity

The Chinese kinship reasoning is much more challenging than English kinship reasoning in the following 2 dimensions:

- **More kinship titles**: The English kinship reasoning benchmark, CLUTRR, contains 24 kinship titles, while our internal Chinese kinship dataset has over 500 possible kinship titles.

---

[5]https://github.com/azreasoners/LLM-ASP

Table 12: Few shot prompt template for StepGame.

Given a story about spatial relations among objects, answer the relation between two queried objects. The answer could only be one of following: [top, bottom_left, top_left, bottom, bottom_right, top_right, right, left, overlap]. If a sentence in the story is describing clock-wise information, then 12 denotes above, 1 and 2 denote upper-right, 3 denotes right, 4 and 5 denote lower-right, 6 denotes below, 7 and 8 denote lower-left, 9 denote left, 10 and 11 denote upper-left. If the sentence is describing cardinal directions, then north denotes above, east denotes right, south denotes below, and west denotes left. Wrap your final answer in brackets. Example: [top].

Story: J is over there and D is on the top of it. S is upper right to W. J is directly south west of S. M is below P and to the right of P. C is sitting at the 3:00 position to D. A is diagonally above D to the left at a 45 degree angle. C is sitting at the 9:00 position of Y. S presents left to Y. J is on the right side to V. What is the relation of the agent A to the agent S?
Answer: [top_left]

Story: Object Y is below object X and to the left of it, too. H is to the right of M. Y is placed at the bottom of U. H is over there and T is on the right. J is directly below V. U is over there and A is on the right of it. U is over there and H is on the right. F is sitting in the left direction of H. M is positioned below Y. What is the relation of the agent X to the agent U?
Answer: [right]

Story: B is to the right of L and is on the same horizontal plane. M and L are next to each other with L on the right and M on the left. B is at the bottom and D is on the top. J is to the top of W vertically. A is to the bottom-left of I. J is sitting at the top position to M. H is above J with a small gap between them. B is on the same horizontal plane directly right to E. E is on the right and W is on the left. What is the relation of the agent L to the agent H?
Answer: [bottom_right]

Story: H and K are side by side with K at the bottom and H on the top. P is below K with a small gap between them. U is there and Z is at the 10 position of a clock face. Object A is above object M and to the right of it, too. D is to the right of H horizontally. P and C are parallel, and P is to the right of C. G and C are vertical and G is above C. Q and E are next to each other with Q on the left and E on the right. The object O is positioned below and to the right of the object J. E is above S at 2 o'clock. F and J are both there with the object F is to the right of object J. Z is over there and N is on the left. Y is diagonally left and below L. If U is the center of a clock face, G is located between 10 and 11. F is directly above W. P is directly north west of V. S is there and L is at the 10 position of a clock face. Q is positioned below D. N is to the bottom left of D. If A is the center of a clock face, Q is located between 4 and 5. What is the relation of the agent E to the agent Z?
Answer: [right]

Story: {story}
Answer:

- **Complex deduction rules**: English kinship treats the maternal family and paternal family in the same way, while Chinese kinship distinguishes between them. The simplest example would be that in English, one's father's brother and mother's brother are both addressed as 'uncle', while in Chinese, they would be titled 'BoBo' and 'JiuJiu', respectively.

It is nearly infeasible to build an ASP program that completely covers such a large label set with these complex rules. Therefore, we did not test ASP in our Chinese kinship experiments.

Table 13: Few shot prompt template for CLUTRR.

Given a story about kinship relations among persons, answer the relation between two queried persons. The answer could only be one of following: [son, grandmother, daughter-in-law, grandson, greatgrandson, grandfather, mother-in-law, greatgranddaughter, uncle, son-in-law, wife, greatgrandfather, brother, husband, daughter, father-in-law, sister, greatgrandmother, granddaughter, aunt, nephew, niece, mother, father].
Story: Edd took his sister Marion out to lunch after learning that she got accepted into her first choice for university. Washington bought to dress for his father Edd Washington and his uncle Bird went to the movies Sunday after church and got popcorn and candy while they were there. What should Marion address Bird?
Answer: [brother]

Story: Ottilia asked her husband Friend if he could chop up some vegetables for dinner. Christine's mother Ottilia was teaching her how to teach when Christine's husband Rollie arrived home. What should Friend address Rollie?
Answer: [son-in-law]
Story: May joined her husband Young, her son Miles and daughter-in-law Abbie for brunch last Sunday. May fixed her husband Young dinner and then they watched a movie they rented. What should Young address Abbie?
Answer: [daughter-in-law]

Story: Leonard and his wife, Ella, went over to Genevieve's house for the weekend. Genevieve told her mother, Ella, that Rose would be over later. Leonard, Rose's father, was happy to hear this. Leila brought her grandmother, Genevieve, some muffins. What should Rose address Genevieve?
Answer: [sister]

Story: {story}
Answer:

Table 14: Few shot prompt template for SPARTUN.

Given a story about spatial relations among objects, answer the relation between two queried objects step by step. The answer could only be one of following: [far, in, touch, has, covered_by, right, overlap, front, behind, cover, left, disconnected_from, below, above, near]. 'inside and touching' refers 'covered_by'. 'inside' and 'within' and 'inside' refers 'in'. 'contain' refers 'has'. If the sentence is describing clock-wise information, then 3 denotes right, 6 denotes below, 9 denotes left, and 12 denotes above. If the sentence is describing cardinal directions, then north denotes above, east denotes right, south denotes below, and west denotes left. There could be multiple answers. Wrap all your answers in brackets. Example: [above, behind].

Story: A box called one covers a medium green apple. Covered by another box called two there is this box. Box two has a medium orange apple which touches a yellow apple. Box two covers the yellow fruit. Where is box two regarding box one?
Answer: [cover]

Story: A midsize orange rectangle is inside and touching a box named DDD. Above and in front of box DDD is another box named EEE. Box DDD is disconnected from and near to this box. A midsize orange rectangle is over and touches another midsize orange rectangle. Midsize orange rectangle number one is within box EEE. Box EEE covers midsize orange rectangle number two. Where is DDD relative to midsize orange rectangle number two?
Answer: [behind, below].

Story: A medium triangle, a big black square and a big circle are in a block called AAA. The big black square is behind the big circle and is in front of the medium triangle. In front of and touches a small black triangle there is this thing. Block AAA has the small black triangle. This block has a small blue square. Behind the medium triangle there is the small black triangle. Behind the big circle is the medium object. The small blue square is in front of the object which was in front of the medium thing. Under the big circle is this shape. What is the position of the medium object regarding the small blue square?
Answer: [behind].

Story: {story}
Answer:

Table 15: Few shot prompt template for Chinese kinship.

You are given a question about chinese kinship relations, please answer the question step by step. ansaers include but not limited to chinese kinship titles: (从/表/堂)侄子,侄女,女婿,儿媳,岳父,岳母,妹夫,姐夫,伯公,叔公,etc. Wrap your final answer in square brackets []. If more than one relation is correct, separate the relations by comma, like: [舅表哥,舅表弟].

Question: 小北最近对家谱的研究产生浓厚兴趣，在整理家族关系时，他发现自己的孙子小明有一位姑妈名叫小花，而小花的奶奶小丽是小北祖辈亲属中的一员。在一次家族聚会上，小北得知他的一个远房亲戚小顾，实际上是小丽的弟弟。根据上述信息，小北可能称呼小顾为？
Answer: [舅舅]。

Question: 在小采的生日宴会上，家人们欢聚一堂，庆祝气氛热烈。小采是一位乐于助人的男性，他总是喜欢带领家人们一起参与各种社会活动。当天，小采的女儿小美邀请了她的嫂子小丽一同参加宴会。小丽和她的老公小帅也都到场了。在聊天中，大家提到了小伶，她是小帅的母亲。在这个喜庆的场合，小采可能称呼小伶为？
Answer: [妻子]。

Question: 小孟的女儿小郁忙着与来宾们打招呼，而小孟则在一旁和她的老公小华交谈。小华提到了他的父亲小闵也即将到来。那么小郁可能称呼小闵为什么？
Answer: [爷爷]。

Question: {story}
Answer:

Table 16: CoT and CoT-SC prompt template for StepGame.

Given a story about spatial relations among objects, answer the relation between two queried objects step by step. The answer could only be one of following: [top, bottom_left, top_left, bottom, bottom_right, top_right, right, left, overlap]. If a sentence in the story is describing clock-wise information, then 12 denotes above, 1 and 2 denote upper-right, 3 denotes right, 4 and 5 denote lower-right, 6 denotes below, 7 and 8 denote lower-left, 9 denote left, 10 and 11 denote upper-left. If the sentence is describing cardinal directions, then north denotes above, east denotes right, south denotes below, and west denotes left. Wrap your final answer in brackets. Example: [top]. A 'Reasoning chain' with respect to the question is provided to help solve the question. Please take it as the important reference. If the 'Reasoning chain' is None, then extact the reasoning chain yourself.

Story: J is over there and D is on the top of it. S is upper right to W. J is directly south west of S. M is below P and to the right of P. C is sitting at the 3:00 position to D. A is diagonally above D to the left at a 45 degree angle. C is sitting at the 9:00 position of Y. S presents left to Y. J is on the right side to V. What is the relation of the agent A to the agent S?
Answer: To determine the relation of agent A to agent S, let's analyze the story step by step:
1. **J is over there and D is on the top of it.** - D is top of J.
2. **S is upper right to W.** - S is top_right of W.
3. **J is directly south west of S.** - J is bottom_left of S.
4. **M is below P and to the right of P.** - M is bottom_right of P.
5. **C is sitting at the 3:00 position to D.** - C is right of D.
6. **A is diagonally above D to the left at a 45 degree angle.** - A is top_left of D.
7. **C is sitting at the 9:00 position of Y.** - C is left of Y.
8. **S presents left to Y.** - S is left of Y.
9. **J is on the right side to V.** - J is right of V.
Now, we need to find the relation between agent A and agent S:
- From step 3, we know J is bottom_left of S. - From step 1, we know D is top of J. - Therefore, D is top of J, and J is bottom_left of S, so D is to the left of S. - From step 6, we know A is top_left of D.
Since D is to the left of S, and A is top_left of D, we can determine that A is top_left of S.
Final answer: [top_left]

Story: Object Y is below object X and to the left of it, too. H is to the right of M. Y is placed at the bottom of U. H is over there and T is on the right. J is directly below V. U is over there and A is on the right of it. U is over there and H is on the right. F is sitting in the left direction of H. M is positioned below Y. What is the relation of the agent X to the agent U?
Answer: To determine the relation of the agent X to the agent U, we need to analyze the spatial relations described in the story step by step:
1. "Object Y is below object X and to the left of it, too." - Y is bottom_left of X.
2. "H is to the right of M." - H is right of M.
3. "Y is placed at the bottom of U." - Y is bottom of U.
4. "H is over there and T is on the right." - T is right of H.
5. "J is directly below V." - J is bottom of V.
6. "U is over there and A is on the right of it." - A is right of U.
7. "U is over there and H is on the right." - H is right of U.
8. "F is sitting in the left direction of H." - F is left of H.
9. "M is positioned below Y." - M is bottom of Y.
Now, let's determine the relation of X to U:
- From step 1, Y is bottom_left of X. - From step 3, Y is bottom of U.
Since Y is bottom of U and bottom_left of X, X must be top_right of U.
Therefore, the relation of the agent X to the agent U is [right].

Story: B is to the right of L and is on the same horizontal plane. M and L are next to each other with L on the right and M on the left. B is at the bottom and D is on the top. J is to the top of W vertically. A is to the bottom-left of I. J is sitting at the top position to M. H is above J with a small gap between them. B is on the same horizontal plane directly right to E. E is on the right and W is on the left. What is the relation of the agent L to the agent H?
Answer: To determine the relation of the agent L to the agent H, let's break down the spatial relations step by step:

Table 17: CoT and CoT-SC prompt template for StepGame. (Continue)

1. **B is to the right of L and is on the same horizontal plane.** - L is to the left of B.
2. **M and L are next to each other with L on the right and M on the left.** - M is to the left of L.
3. **B is at the bottom and D is on the top.** - B is below D.
4. **J is to the top of W vertically.** - J is above W.
5. **A is to the bottom-left of I.** - A is bottom_left of I.
6. **J is sitting at the top position to M.** - J is above M.
7. **H is above J with a small gap between them.** - H is above J.
8. **B is on the same horizontal plane directly right to E.** - E is to the left of B.
9. **E is on the right and W is on the left.** - W is to the left of E.
Now, let's find the relation between L and H:
- From steps 1 and 2, we know that M is to the left of L, and L is to the left of B. - From step 6, J is above M. - From step 7, H is above J.
Since M is to the left of L, and J is above M, J is also to the left of L but higher up. Since H is above J, H is even higher up and still to the left of L.
Therefore, the relation of L to H is [bottom_right].

Story: H and K are side by side with K at the bottom and H on the top. P is below K with a small gap between them. U is there and Z is at the 10 position of a clock face. Object A is above object M and to the right of it, too. D is to the right of H horizontally. P and C are parallel, and P is to the right of C. G and C are vertical and G is above C. Q and E are next to each other with Q on the left and E on the right. The object O is positioned below and to the right of the object J. E is above S at 2 o'clock. F and J are both there with the object F is to the right of object J. Z is over there and N is on the left. Y is diagonally left and below L. If U is the center of a clock face, G is located between 10 and 11. F is directly above W. P is directly north west of V. S is there and L is at the 10 position of a clock face. Q is positioned below D. N is to the bottom left of D. If A is the center of a clock face, Q is located between 4 and 5. What is the relation of the agent E to the agent Z?
Answer: To determine the spatial relation between agent E and agent Z, let's break down the information provided in the story step by step:
1. **Position of Z:** - "Z is at the 10 position of a clock face." - If U is the center of a clock face, "G is located between 10 and 11." - "Z is over there and N is on the left." - "S is there and L is at the 10 position of a clock face."
From this, we can infer that Z is at the 10 o'clock position relative to U.
2. **Position of E:** - "Q and E are next to each other with Q on the left and E on the right." - "E is above S at 2 o'clock." - "If A is the center of a clock face, Q is located between 4 and 5."
From this, we can infer that E is at the 2 o'clock position relative to S.
3. **Relative Positions:** - We know Z is at the 10 o'clock position relative to U. - We know E is at the 2 o'clock position relative to S.
4. **Combining Information:** - Since Z is at the 10 o'clock position relative to U, Z is in the upper-left direction. - Since E is at the 2 o'clock position relative to S, E is in the upper-right direction.
To find the relation of E to Z, we need to consider their relative positions: - Z is at the 10 o'clock position (upper-left). - E is at the 2 o'clock position (upper-right).
Therefore, E is to the **right** of Z.
Final answer: [right]

Story: {input}
Answer:

Table 18: CoT and CoT-SC prompt template for CLUTRR.

Given a story about kinship relations among persons, answer the relation between two queried persons step by step. The answer could only be one of following: [aunt, son-in-law, grandfather, brother, sister, father, mother, grandmother, uncle, daughter-in-law, grandson, granddaughter, father-in-law, mother-in-law, nephew, son, daughter, niece, husband, wife, sister-in-law]. Wrap your final answer in brackets. Example: [grandfather].

Story: Edd took his sister Marion out to lunch after learning that she got accepted into her first choice for university. Washington bought to dress for his father Edd. Washington and his uncle Bird went to the movies Sunday after church and got popcorn and candy while they were there. How should Marion address Bird?
Answer: To determine how Marion should address Bird, we need to carefully analyze the relationships mentioned in the story.
1. Edd took his sister Marion out to lunch. - This tells us that Marion is Edd's sister.
2. Washington bought a dress for his father Edd. - This tells us that Washington is Edd's son, making Edd Washington's father.
3. Washington and his uncle Bird went to the movies. - This tells us that Bird is Washington's uncle.
Since Bird is Washington's uncle and Washington is Edd's son, Bird must be Edd's brother. Therefore, Bird is Marion's brother, because Marion is Edd's sister.
So, Marion should address Bird as her brother.
Final answer: [brother]

Story: Ottilia asked her husband Friend if he could chop up some vegetables for dinner. Christine's mother Ottilia was teaching her how to teach when Christine's husband Rollie arrived home. How should Friend address Rollie?
Answer: To determine how Friend should address Rollie, we need to understand the relationships between the characters in the story.
1. Ottilia asked her husband Friend to chop vegetables. - Ottilia and Friend are married.
2. Christine's mother is Ottilia. - Christine is Ottilia's daughter.
3. Christine's husband is Rollie. - Rollie is Christine's husband.
Since Ottilia and Friend are married, and Ottilia is Christine's mother, Friend is Christine's father. Therefore, Rollie, being Christine's husband, is Friend's son-in-law.
So, Friend should address Rollie as his son-in-law.
Final answer: [son-in-law]

Story: May joined her husband Young, her son Miles and daughter-in-law Abbie for brunch last Sunday. May fixed her husband Young dinner and then they watched a movie they rented. How should Young address Abbie?
Answer: To determine how Young should address Abbie, we need to establish their relationship based on the information given in the story.
1. Identify the relationships: - May is Young's wife. - Miles is May and Young's son. - Abbie is Miles' wife, making her May and Young's daughter-in-law.
2. Since Abbie is married to Miles, Young should address Abbie as his daughter-in-law.
Therefore, the appropriate way for Young to address Abbie is:
[daughter-in-law]

Story: Leonard and his wife, Ella, went over to Genevieve's house for the weekend. Genevieve told her mother, Ella, that Rose would be over later. Leonard, Rose's father, was happy to hear this. Leila brought her grandmother, Genevieve, some muffins. How should Rose address Genevieve?
Answer: To determine how Rose should address Genevieve, let's break down the relationships step by step:
1. Leonard and Ella are married. 2. Genevieve is Ella's daughter, making Genevieve Leonard's daughter as well.
3. Rose is Leonard's daughter, which means Rose is also Ella's daughter. 4. Since Genevieve is Ella and Leonard's daughter, Genevieve is Rose's sister.
Given these relationships, Rose should address Genevieve as her sister.
Final answer: [sister]

Story: {story}
Answer:

Table 19: CoT and CoT-SC prompt templates for SPARTUN.

Given a story about spatial relations among objects, answer the relation between two queried objects step by step. The answer could only be one of following: [far, in, touch, has, covered_by, right, overlap, front, behind, cover, left, disconnected_from, below, above, near]. 'inside and touching' refers 'covered_by'. 'inside' and 'within' and 'inside' refers 'in'. 'contain' refers 'has'. If the sentence is describing clock-wise information, then 3 denotes right, 6 denotes below, 9 denotes left, and 12 denotes above. If the sentence is describing cardinal directions, then north denotes above, east denotes right, south denotes below, and west denotes left. There could be multiple answers. Wrap all your answers in brackets. Example: [above, behind].

Story: A box called one covers a medium green apple. Covered by another box called two there is this box. Box two has a medium orange apple which touches a yellow apple. Box two covers the yellow fruit. Where is box two regarding box one? Answer: To determine the spatial relation between box two and box one, we need to analyze the given story step by step.
1. **First Sentence:** "A box called one covers a medium green apple." - Box one is covering the green apple.
2. **Second Sentence:** "Covered by another box called two there is this box." - Box one is covered by box two.
From these two sentences, we can infer the following: - Box one is covered by box two.
Therefore, the spatial relation between box two and box one is that box two is covering box one.
Final answer: [cover]

Story: A midsize orange rectangle is inside and touching a box named DDD. Above and in front of box DDD is another box named EEE. Box DDD is disconnected from and near to this box. A midsize orange rectangle is over and touches another midsize orange rectangle. Midsize orange rectangle number one is within box EEE. Box EEE covers midsize orange rectangle number two. Where is DDD relative to midsize orange rectangle number two?
Answer: To determine the spatial relation between box DDD and midsize orange rectangle number two, let's break down the given story step by step:
1. "A midsize orange rectangle is inside and touching a box named DDD." - This indicates that there is a midsize orange rectangle within box DDD and touching its sides.
2. "Above and in front of box DDD is another box named EEE." - Box EEE is positioned above and in front of box DDD.
3. "Box DDD is disconnected from and near to this box." - Box DDD is near but not touching box EEE.
4. "A midsize orange rectangle is over and touches another midsize orange rectangle." - There are two midsize orange rectangles, and one is positioned over and touching the other.
5. "Midsize orange rectangle number one is within box EEE." - The first midsize orange rectangle is inside box EEE.
6. "Box EEE covers midsize orange rectangle number two." - Box EEE is covering the second midsize orange rectangle.
From these statements, we can infer the following: - Box EEE is above and in front of box DDD. - The second midsize orange rectangle is covered by box EEE, meaning it is inside or directly beneath box EEE.
Since box EEE is above and in front of box DDD, and box EEE covers the second midsize orange rectangle, it implies that the second midsize orange rectangle is also above and in front of box DDD.
Therefore, the relation between box DDD and midsize orange rectangle number two is that DDD is behind and below the second midsize orange rectangle.
Final answer: [behind, below].

Story: A medium triangle, a big black square and a big circle are in a block called AAA. The big black square is behind the big circle and is in front of the medium triangle. In front of and touches a small black triangle there is this thing. Block AAA has the small black triangle. This block has a small blue square. Behind the medium triangle there is the small black triangle. Behind the big circle is the medium object. The small blue square is in front of the object which was in front of the medium thing. Under the big circle is this shape. What is the position of the medium object regarding the small blue square?
Answer: To determine the position of the medium object regarding the small blue square, let's break down the story step by step:
1. **Objects in Block AAA**: - Medium triangle - Big black square - Big circle - Small black triangle - Small blue square

Table 20: CoT and CoT-SC prompt templates for SPARTUN. (Continue)

2. **Relations**: - The big black square is behind the big circle. - The big black square is in front of the medium triangle. - The small black triangle is touched by something in front of it. - The small black triangle is in Block AAA. - The small blue square is in Block AAA. - The small black triangle is behind the medium triangle. - The medium object (medium triangle) is behind the big circle. - The small blue square is in front of the object which was in front of the medium object (medium triangle). - Under the big circle is another shape (not directly relevant to our query).

3. **Key Relations for Query**: - The medium triangle is behind the big circle. - The big black square is behind the big circle and in front of the medium triangle. - The small blue square is in front of the object which was in front of the medium triangle.

4. **Position of Medium Object**: - The medium triangle is behind the big circle. - The small blue square is in front of the object which was in front of the medium triangle. Since the big black square is in front of the medium triangle, the small blue square must be in front of the big black square.

Given these relations, the medium triangle (medium object) is behind both the big black square and the small blue square.

Therefore, the position of the medium object regarding the small blue square is [behind].

Story: {story}
Answer:

Table 21: CoT and CoT-SC prompt templates for Chinese kinship.

You are given a question about chinese kinship relations, please answer the question step by step. ansaers include but not limited to chinese kinship titles: (从/表/堂)侄子,侄女,女婿,儿媳,岳父,岳母,妹夫,姐夫,伯公,叔公,etc. Wrap your final answer in square brackets []. If more than one relation is correct, separate the relations by comma, like: [舅表哥,舅表弟].

Question: 小北最近对家谱的研究产生浓厚兴趣，在整理家族关系时，他发现自己的孙子小明有一位姑妈名叫小花，而小花的奶奶小丽是小北祖辈亲属中的一员。在一次家族聚会上，小北得知他的一个远房亲戚小顾，实际上是小丽的弟弟。根据上述信息，小北可能称呼小顾为?
Answer: 根据题目提供的信息，我们可以逐步推导出小北与小顾的关系:
1. 小北的孙子是小明。 2. 小明有一位姑妈名叫小花。 3. 小花的奶奶是小丽。 4. 小丽是小北祖辈亲属中的一员。
首先，明确小花的身份: - 小花是小北的孙子小明的姑妈，说明小花是小北的女儿。
接下来，明确小丽的身份: - 小花的奶奶是小丽，说明小丽是小北的母亲。
然后，明确小顾的身份: - 小顾是小丽的弟弟，说明小顾是小北的舅舅。
因此，小北可能称呼小顾为[舅舅]。

Question: 在小采的生日宴会上，家人们欢聚一堂，庆祝气氛热烈。小采是一位乐于助人的男性，他总是喜欢带领家人们一起参与各种社会活动。当天，小采的女儿小美邀请了她的嫂子小丽一同参加宴会。小丽和她的老公小帅也都到场了。在聊天中，大家提到了小伶，她是小帅的母亲。在这个喜庆的场合，小采可能称呼小伶为?
Answer: To determine how 小采 might refer to 小伶, we need to establish the relationships step by step:
1. 小采 has a daughter named 小美. 2. 小美 invited her 嫂子 (sister-in-law) 小丽 to the birthday party. 3. 小丽's husband is 小帅. 4. 小帅 is 小采's son because 小丽 is referred to as 小美's 嫂子, meaning 小帅 is 小美's brother. 5. 小伶 is 小帅's mother.
Since 小帅 is 小采's son, 小伶 must be 小采's wife.
Therefore, 小采 would refer to 小伶 as [妻子].

Question: 小孟的女儿小郁忙着与来宾们打招呼，而小孟则在一旁和她的老公小华交谈。小华提到了他的父亲小闵也即将到来。那么小郁可能称呼小闵为什么?
Answer: To determine what 小郁 (Xiao Yu) would call 小闵 (Xiao Min), we need to understand the relationships described in the question:
1. 小孟 (Xiao Meng) is the mother of 小郁 (Xiao Yu). 2. 小孟's husband is 小华 (Xiao Hua), making 小华 the father of 小郁. 3. 小华 mentioned his father 小闵 (Xiao Min).
Thus, 小闵 (Xiao Min) is 小郁's (Xiao Yu's) paternal grandfather.
In Chinese kinship terms, a paternal grandfather is called 祖父 or 爷爷.
Therefore, 小郁 would call 小闵:
[爷爷]

Question: {story}
Answer:

Table 22: LLM-ASP extraction prompt template for StepGame. The prompt are a slightly modified version of the original so that all triplets and queries are extracted at once.

Given a story, please parse each sentence into a fact. If the sentence is describing clock-wise information, then 12 denotes top, 1 and 2 denote top_right, 3 denotes right, 4 and 5 denote down_right, 6 denotes down, 7 and 8 denote down_left, 9 denote left, 10 and 11 denote top_left. If the sentence is describing cardinal directions, then north denotes top, east denotes right, south denotes down, and west denotes left. If the sentence is a question, the fact starts with query. Otherwise, the fact starts with one of top, down, left, right, top_left, top_right, down_left, and down_right.

Story: If H is the center of a clock face, X is located between 4 and 5. V is directly north east of D. H and E are next to each other with H on the left and E on the right. What is the relation of the agent H to the agent E? Semantic Parse: top_left("H", "X"). top_right("V", "D"). left("H", "E"). query("H", "E").

Story: I and P are parallel, and I on the right of P. K is above I and to the right of I. B and P are parallel, and B is to the right of P. P is below J with a small gap between them. T is below A at 7 o'clock. What is the relation of the agent I to the agent B? Semantic Parse: right("I", "P"). top_right("K", "I"). right("B", "P"). down("P", "J"). down_left("T", "A"). query("I", "B").

Story: Z is below S with a small gap between them. The object M is positioned directly below the object J. A is on the left side of and below M. Y presents upper right to N. B is positioned down and to the left of M. N is over there and C is on the right. W and A are parallel, and W on the left of A. S and D are both there with the object S is to the right of object D. W is at the bottom of D. Z is at W's 9 o'clock. What is the relation of the agent A to the agent M? Semantic Parse: down("Z", "S"). down("M", "J"). down_left("A", "M"). top_right("Y", "N"). down_left("B", "M"). right("C", "N"). left("W", "A"). right("S", "D"). down("W", "D"). left("Z", "W"). query("A", "M").

Story: H and Y are in a horizontal line with H on the left. V is at the 6 o'clock position relative to X. The object U is positioned below and to the right of the object W. R is diagonally left and below D. Z presents below I. Z is diagonally above P to the right at a 45 degree. Object P is above object R and to the left of it, too. I is placed on the top of V. N is positioned up and to the right of D. X is at Z's 6 o'clock. Y is over there and V is at the bottom of it. What is the relation of the agent N to the agent X? Semantic Parse: left("H", "Y"). down("V", "X"). down_right("U", "W"). down_left("R", "D"). down("Z", "I"). top_right("Z", "P"). top_left("P", "R"). top("I", "V"). top_right("N", "D"). down("X", "Z"). down("V", "Y"). query("N", "X").

Story: H and K are side by side with K at the bottom and H on the top. P is below K with a small gap between them. U is there and Z is at the 10 position of a clock face. Object A is above object M and to the right of it, too. D is to the right of H horizontally. P and C are parallel, and P is to the right of C. G and C are vertical and G is above C. Q and E are next to each other with Q on the left and E on the right. The object O is positioned below and to the right of the object J. E is above S at 2 o'clock. F and J are both there with the object F is to the right of object J. Z is over there and N is on the left. Y is diagonally left and below L. If U is the center of a clock face, G is located between 10 and 11. F is directly above W. P is directly north west of V. S is there and L is at the 10 position of a clock face. Q is positioned below D. N is to the bottom left of D. If A is the center of a clock face, Q is located between 4 and 5. What is the relation of the agent E to the agent Z? Semantic Parse: down("K", "H"). down("P", "K"). top_left("Z", "U"). top_right("A", "M"). right("D", "H"). right("P", "C"). top("G", "C"). left("Q", "E"). down_right("O", "J"). top_right("E", "S"). right("F", "J"). left("N", "Z"). down_left("Y", "L"). top_left("G", "U"). top("F", "W"). top_left("P", "V"). top_left("L", "S"). down("Q", "D"). down_left("N", "D"). down_right("Q", "A"). query("E", "Z").

Story: {story}

Table 23: LLM-ASP relation extraction prompt template for CLUTRR. The prompt are a slightly modified version of the original so that all triplets and queries are extracted at once.

Given a story, extract atomic facts of the form relation("Person", "Person"). Example relations are: father, mother, parent, son, daughter, child, grandfather, grandmother, grandson, granddaughter, wife, husband, spouse, sibling, nephew, niece, uncle, aunt, child_in_law, and parent_in_law. Do not answer the query.
Story: Edd took his sister Marion out to lunch after learning that she got accepted into her first choice for university. Washington bought to dress for his father Edd. Washington and his uncle Bird went to the movies Sunday after church and got popcorn and candy while they were there. What should Marion address Bird? Semantic Parse: sister("Edd","Marion"). father("Washington", "Edd"). uncle("Washington", "Bird"). query("Marion", "Bird").
Story: Michelle was excited for today, its her daughter's, Theresa, spring break. She will finally get to see her. Michael was busy and sent his wife, Marlene, instead. Kristen loved to care for her newborn child Ronald. Eric's son is Arthur. What should Theresa address Michelle? Semantic Parse: daughter("Michelle", "Theresa"). wife("Michael", "Marlene"). child("Kristen", "Ronald"). son("Eric", "Arthur"). query("Theresa", "Michelle").
Story: Vernon was present in the delivery room when his daughter Raquel was born, but when his daughter Constance was born he was too sick. Vernon and his daughter Margaret went to the movies. Constance, Margaret's sister, had to stay home as she was sick. What should Raquel address Margaret? Semantic Parse: daughter("Vernon", "Raquel"). daughter("Vernon", "Constance"). daughter("Vernon", "Margaret"). sister("Margaret", "Constance"). query("Raquel", "Margaret").
Story: Eric who is Carl's father grounded Carl after finding out what Carl had done at school. Ronald was busy planning a 90th birthday party for his aunt, Theresa. Eric and his son Carl went to the park and saw Eric's father Kyle there with his dog. What should Carl address Kyle? Semantic Parse: father("Carl", "Eric"). aunt("Ronald", "Theresa"). son("Eric", "Carl"). father("Eric", "Kyle"). query("Carl", "Kyle").
Story: Shirley and Edward are siblings and best friends. They do everything together. Henry walked his daughters Amanda and Michelle to school. Kyle enjoys watching movies with his son's daughter. Her name is Amanda. What should Kyle address Michelle? Semantic Parse: sibling("Shirley", "Edward"). daughter("Henry", "Amanda"). daughter("Henry", "Michelle"). granddaughter("Kyle", "Amanda"). query("Kyle", "Michelle").
Story: Michael is taking his wife Henry out to dinner for their date tonight. Avis went with her grandmother, Henry, to the grocery store to help her while she shopped. Alberta, who is the sister of Avis, is a lovely girl. What should Michael address Alberta? Semantic Parse: wife("Michael", "Henry"). grandmother("Avis", "Henry"). sister("Avis", "Alberta"). query("Michael", "Alberta").
Story: Allen's father, Eric, bought him some ice cream. Karen was baking cookies for her grandson, Allen. Allen's brother Arthur came home from school, so she baked some extra for him, too. Eric's son, Arthur, was ill and needed to be picked up at school. Eric hurried to his side. What should Karen address Arthur? Semantic Parse: father("Allen", "Eric"). grandson("Karen", "Allen"). brother("Allen", "Arthur"). son("Eric", "Arthur"). query("Karen", "Arthur").
Story: Karen was spending the weekend with her grandson, Eddie. Eddie's sister Michelle was supposed to come too, but she was busy and could n't make it. Theresa took her daughter, Michelle, out to High Tea yesterday afternoon. Eddie's mother Theresa baked brownies for dessert after they had dinner. What should Karen address Michelle? Semantic Parse: grandson("Karen", "Eddie"). sister("Eddie", "Michelle"). daughter("Theresa", "Michelle"). mother("Eddie", "Theresa"). query("Karen", "Michelle").

Story: {story}

Table 24: LLM-ASP gender extraction prompt template for CLUTRR. The prompt template are a slightly modified version of the original so that all genders are extracted at once.

Given a story, extract atomic facts of the form male("Person") or female("Person") for every person that appears in the sentences.

Story: Edd took his sister Marion out to lunch after learning that she got accepted into her first choice for university. Washington bought to dress for his father Edd. Washington and his uncle Bird went to the movies Sunday after church and got popcorn and candy while they were there. What should Marion address Bird? Semantic Parse: male("Edd"). female("Marion"). male("Washington"). male("Bird").

Story: Michelle was excited for today, its her daughter's, Theresa, spring break. She will finally get to see her. Michael was busy and sent his wife, Marlene, instead. Kristen loved to care for her newborn child Ronald. Eric's son is Arthur. What should Theresa address Michelle? Semantic Parse: female("Michelle"). female("Theresa"). male("Michael"). female("Marlene"). female("Kristen"). male("Ronald"). male("Eric"). male("Arthur").

Story: Vernon was present in the delivery room when his daughter Raquel was born, but when his daughter Constance was born he was too sick. Vernon and his daughter Margaret went to the movies. Constance, Margaret's sister, had to stay home as she was sick. What should Raquel address Margaret? Semantic Parse: male("Vernon"). female("Raquel"). female("Constance"). female("Margaret").

Story: Eric who is Carl's father grounded Carl after finding out what Carl had done at school. Ronald was busy planning a 90th birthday party for his aunt, Theresa. Eric and his son Carl went to the park and saw Eric's father Kyle there with his dog. What should Carl address Kyle? Semantic Parse: male("Eric"). male("Carl"). male("Ronald"). female("Theresa"). male("Kyle").

Story: Shirley and Edward are siblings and best friends. They do everything together. Henry walked his daughters Amanda and Michelle to school. Kyle enjoys watching movies with his son's daughter. Her name is Amanda. What should Kyle address Michelle? Semantic Parse: female("Shirley"). male("Edward"). male("Henry"). female("Amanda"). female("Michelle"). male("Kyle").

Story: Michael is taking his wife Henry out to dinner for their date tonight. Avis went with her grandmother, Henry, to the grocery store to help her while she shopped. Alberta, who is the sister of Avis, is a lovely girl. What should Michael address Alberta? Semantic Parse: male("Michael"). female("Henry"). female("Avis"). female("Alberta").

Story: Allen's father, Eric, bought him some ice cream. Karen was baking cookies for her grandson, Allen. Allen's brother Arthur came home from school, so she baked some extra for him, too. Eric's son, Arthur, was ill and needed to be picked up at school. Eric hurried to his side. What should Karen address Arthur? Semantic Parse: male("Allen"). male("Eric"). female("Karen"). male("Arthur").

Story: Karen was spending the weekend with her grandson, Eddie. Eddie's sister Michelle was supposed to come too, but she was busy and could n't make it. Theresa took her daughter, Michelle, out to High Tea yesterday afternoon. Eddie's mother Theresa baked brownies for dessert after they had dinner. What should Karen address Michelle? Semantic Parse: female("Karen"). male("Eddie"). female("Michelle"). female("Theresa").

Story: {story}

Table 25: Extraction mistakes for CLUTRR dataset.

| Type | Sentence | Extracted Triplet |
| --- | --- | --- |
| Wrong Relation | Elsie and Lewis did the Father daughter dance at the recital and his wife Dollie was so happy she cried | [Elsie<female>, husband, Lewis<male>] |
| Wrong Source | Maynard and his son Dana went to his mother Corine's home Dana received a novel for Christmas from his aunt Lou. | [Dana<male>, mother, Corine<female>] |
| Wrong Gender | Friend's mother Ottilia had to help him with his homework because he was having a test soon. | [Friend<unknown>, mother, Ottilia<female>] |
| Wrong Relation | Clarence has 3 children, and one grandson. The Grandsons name is Tony | [Clarence<unknown>, children, Tony<male>] |

Table 26: Reasoning errors on CLUTRR dataset.

| Reasoning | Wrong reasoning |
| --- | --- |
| Nicolas is Clarence's grandson. Nicolas is June's sister. Since Nicolas is Clarence's grandson, this means June is Clarence's daughter. | Wrong relation deduction |
| Jacob is Lorraine's brand-new baby brother. Clarence should address Jacob as his [grandchild]. | Fail to specify the grandson. Given information that Jacob is male |

Table 27: The full relation extraction prompt template used for the StepGame extraction that demonstrates our prompting methods: Structured Prompts with Sectional Markup , Structured Output with Syntactic Delimiters , Predefined Output Categories , Decomposition of the Extraction Task

---

\# Background information
Given a story about spatial relations among agents and finally a query asking about the relation between two agents. Please extract triplets encoding the relations between two agents as well as the query.
\# Predefined relations
Possible relations are: top, down, left, right, top_left, top_right, down_left, and down_right.
\# Hints
- If a sentence in the story is describing clock-wise information, then 12 denotes above, 1 and 2 denote top_right, 3 denotes right, 4 and 5 denote down_right, 6 denotes down, 7 and 8 denote down_left, 9 denote left, 10 and 11 denote top_left.
- If the sentence is describing cardinal directions, then north denotes top, east denotes right, south denotes down, and west denotes left.
- Note that front means top; above and right means top_right; below and left means down_left, etc.
\# Output format
Write each triplet on a new line. The triplet should be in the format: [( A,relation,B )] ; the query should be in the format: [( A,B )] , i.e., you should use nothing but a single letter to represent an agent. Do not output thinking process.
\# EXAMPLE
- STORY: H and K are side by side with K at the bottom and H on the top.
P is below K with a small gap between them.
U is there and Z is at the 10 position of a clock face.
......
What is the relation of the agent E to the agent Z?
- RELATIONSHIP :
[( H,top,K )] ,
[( P,down,K )] ,
[( Z,top_left,U )] ,
......
- QUERY :
[( E,Z )]
Please fill in RELATIONSHIP and QUERY.
- STORY:input
- RELATIONSHIP : [ FILL_IN ]
- QUERY : [ FILL_IN ]

Table 28: The full relation extraction prompt template used for the CLUTRR extraction that demonstrates our prompting methods: Structured Prompts with Sectional Markup, Structured Output with Syntactic Delimiters, Predefined Output Categories, Decomposition of the Extraction Task

---

\# Placeholders in the triplets:

- relation_query : kinship in question of the input.

- label the gender of person by: < male > , < female > , and < unknown >

\# Explanation of sections

- STORY: contains kinship keywords between the characters.

- RELATIONSHIP: summarize the kinship relations with triplets, with every triplet represent the kinship of 2 characters. For example, ( Terry < male > ,daughter,Mozella < female > ) means "Terry's daughter is Mozella", or "Mozella is the daughter of Terry". You should label every character with < male > , < female > , or < unknown > if the gender is uncertain.

- QUERY: the final question about a kinship, also represented by triplets. For example, if the question asks about how should A < male > addresses B < female > , the triplet should be ( A < male > ,relation_query,B < female > )

\# Examples

\#\# Example 1

- STORY: 'Edd took his sister Marion out to lunch after learning that she got accepted into her first choice for university. Washington bought to dress for his father Edd. Washington and his uncle Bird went to the movies Sunday after church and got popcorn and candy while they were there. What should Marion address Bird?'

- RELATIONSHIP :

[( Edd < male > ,sister,Marion < female > ) ,

( Washington < male > ,father,Edd < male > ) ,

( Washington < male > ,uncle,Bird < male > )]

- QUERY :

[( Marion < female > ,relation_query,Bird < male > )]

\#\# Example 2

......

Please fill in the sections: RELATIONSHIP and QUERY of Example 3 below

\#\# Example 3:

- STORY:'input'

- RELATIONSHIP :[ FILL_IN ]

- QUERY :[ FILL_IN ]

---

Table 29: The relation extraction prompt template used for the SPARTUN dataset, which demonstrates our prompting methods: Structured Prompts with Sectional Markup , Structured Output with Syntactic Delimiters , Predefined Output Categories , Decomposition of the Extraction Task

---

A problem consists of a story and a question.

For story, please parse all relations between entities into a list of triplets in the format: [( A,relation,B )] .

For question, please parse the pair of entities asked in the format: [( A,B )] .

Possible relations are: [far, in, touch, has, covered_by, right, overlap, front, behind, cover, left, disconnected_from, below, above, near] .

If the sentence is describing clock-wise information, then 3 denotes right, 6 denotes below, 9 denotes left, and 12 denotes above.

If the sentence is describing cardinal directions, then north denotes above, east denotes right, south denotes below, and west denotes left.

Write each triplet on a new line.

# EXAMPLE

- STORY: A medium triangle, a big black square and a big circle are in a block called AAA. The big black square is behind the big circle and is in front of the medium triangle.

......

- RELATIONSHIP :

[( medium triangle,in,block AAA )] ,

[( big black square,in,block AAA )] ,

[( big circle,in,block AAA )] ,

[( big black square,behind,big circle )] ,

[( big black square,front,medium triangle )] ,

.....

- QUERY :

[( medium triangle,small blue square )]

Please fill in RELATIONSHIP and QUERY.

- STORY: [ 'input' ]

- RELATIONSHIP : [ FILL_IN ]

- QUERY : [ FILL_IN ]

---

Table 30: The relation extraction prompt template used for the Chinese kinship dataset, which demonstrates our prompting methods: Structured Prompts with Sectional Markup, Structured Output with Syntactic Delimiters, Predefined Output Categories, Decomposition of the Extraction Task

# Placeholders in the triplets:
- P1, P2, P3, and so on: alias for the person appeared in the original input.
- person_query: person in question of the input.
- relation_query: kinship in question of the input.
- label the gender of person by: < male >, < female >, and < unknown >
- label the age by: < older >, < younger >, and < unknown >
# Examples
## Example 1
- ORIGINAL_INPUT: '在外婆的80岁庆生宴上，当小明的妈妈指着一位老先生说那是你的姨外祖父时，请问，这位老先生和小明的外婆是什么关系？'
- RELATIONSHIP:
[( P1 < unknown >,外婆 < younger >,P2 < female >),
( P1 < unknown >,妈妈 < younger >,P3 < female >),
( P1 < unknown >,姨外祖父 < younger >,P4 < male > )]
- QUERY:
[( P4 < male >,relation_query < unknown >,P2 < female > )]
## Example 2
......
# Structure of examples
- ORIGINAL_INPUT: contains information of the kinships between the people mentioned
- RELATIONSHIP: summarize the kinships with triplets, with every triplet represent the kinship of 2 people. Include the relative seniority in the middle kinship element. For example, ( P2 < female >,妈妈 < younger >,P3 < female >) means P3 is 妈妈 of P2, or P2的妈妈是P3; P2 is younger than P3.
- QUERY: the final question about a kinship or a person, also represented by triplets. For example, if the question asks about how should P2 < male > addresses P4 < female > and P2 is older than P4, the triplet should be ( P2 < male >,relation_query < older >,P4 < female >)
You should label the relationships in sections RELATIONSHIP and QUERY with the relative age: < older > means "is older than" and < younger > means "is younger than". Note that there might be descriptions about the ages of the people or their relative seniority, like "小红比他小两岁" meaning "小红"" is younger than "他".
Please fill in the sections: RELATIONSHIP, and QUERY of Example 3
## Example 3:
- ORIGINAL_INPUT: [ 'input' ]
- RELATIONSHIP : [ FILL_IN ]
- QUERY : [ FILL_IN ]

Table 31: PoT-LLM prompt template for StepGame.

Given a story about spatial relations among objects, answer the relation between two queried objects step by step. The answer could only be one of following: [top, bottom_left, top_left, bottom, bottom_right, top_right, right, left, overlap]. If a sentence in the story is describing clock-wise information, then 12 denotes above, 1 and 2 denote upper-right, 3 denotes right, 4 and 5 denote lower-right, 6 denotes below, 7 and 8 denote lower-left, 9 denote left, 10 and 11 denote upper-left. If the sentence is describing cardinal directions, then north denotes above, east denotes right, south denotes below, and west denotes left. Wrap your final answer in brackets. Example: [top].

Story: A is at the top left of D, D is at the top of J, J is at the bottom left of S. What is the relation of the A to the S?
Answer: Let's break down the spatial relations step by step:
1. A is at the top left of D. - This means A is positioned at the top left relative to D.
2. D is at the top of J. - This means D is positioned above J.
3. J is at the bottom left of S. - This means J is positioned at the bottom left relative to S.
Now, let's determine the relation of A to S:
- Since A is at the top left of D, and D is at the top of J, A is also above J but slightly to the left. - Since J is at the bottom left of S, and A is above J (and slightly to the left), A will be positioned at the top left relative to S. Thus, the relation of the A to the S is [top_left].

Story: Y is at the bottom left of X, Y is at the bottom of U. What is the relation of the X to the U?
Answer: To determine the relation of X to U, we need to follow the spatial relations step by step as described in the story.
1. Y is at the bottom left of X. - This means X is at the top right of Y.
2. Y is at the bottom of U. - This means U is at the top of Y.
Now, combining these two pieces of information: - Since X is at the top right of Y and U is at the top of Y, we need to determine the relation of X to U.
From the spatial relations: - If X is at the top right of Y and U is at the top of Y, then X is to the right of U. Therefore, the relation of X to U is [right].

Story: L is at the right of M, J is at the top of M, H is at the top of J. What is the relation of the L to the H?
Answer: Let's break down the spatial relations step by step according to the story:
1. **L is at the right of M**: - This means L is positioned to the right of M.
2. **J is at the top of M**: - This means J is positioned above M.
3. **H is at the top of J**: - This means H is positioned above J.
Now, let's visualize the positions of each object:
- M is our reference point. - L is to the right of M. - J is above M. - H is above J.
Since J is directly above M, and H is directly above J, H is also directly above M.
Now, considering the position of L (which is to the right of M) and H (which is above M), we can determine the relation between L and H.
- L is to the right of M. - H is above M.
Therefore, L is positioned to the bottom-right of H.
The final answer is: [bottom_right].

Story: E is at the right of Q, Q is at the bottom of D, N is at the bottom left of D, N is at the left of Z. What is the relation of the E to the Z?
Answer: To determine the spatial relation of E to Z, let's analyze the story step by step:
1. **E is at the right of Q**: - This means E is to the right of Q.
2. **Q is at the bottom of D**: - This means Q is below D.
3. **N is at the bottom left of D**: - This means N is below and to the left of D.
4. **N is at the left of Z**: - This means N is to the left of Z.

Table 32: PoT-LLM prompt template for StepGame. (Continue)

Now let's combine these relations to find the position of E relative to Z:
- Since Q is at the bottom of D, and E is to the right of Q, E is to the right and slightly below D. - N is at the bottom left of D, which means N is below and to the left of D. - N is to the left of Z, which means Z is to the right of N. - Since E is to the right of Q and Q is below D, E is to the right and slightly below D. - Given that N is to the left of Z, and N is at the bottom left of D, Z must be to the right and slightly above N.
Combining all these, E is to the right of Q, which is below D, and Z is to the right of N, which is at the bottom left of D. Therefore, E is to the right and slightly below D, while Z is to the right of N and slightly above N.
Thus, E is to the right and slightly below D, and Z is to the right of N. Since N is at the bottom left of D, Z is to the right of N, making Z to the right and slightly above N. Therefore, E is to the right of Z.
Final answer: [right].

Story: {extacted instance path}
Answer:

Table 33: PoT-LLM prompt template for CLUTRR.

Given a story about kinship relations among persons, answer the relation between two queried persons step by step. The answer could only be one of following: [aunt, son-in-law, grandfather, brother, sister, father, mother, grandmother, uncle, daughter-in-law, grandson, granddaughter, father-in-law, mother-in-law, nephew, son, daughter, niece, husband, wife, sister-in-law]. Wrap your final answer in brackets. Example: [grandfather].
Story: Marion is Edd's sister, Edd is Washington's father, Bird is Washington's uncle. Marion is female, Edd is male, Washington is male, Bird is male. How should Marion address Bird?
Answer: To determine how Marion should address Bird, let's break down the relationships step by step:
1. Marion is Edd's sister. 2. Edd is Washington's father. 3. Washington is male. 4. Bird is Washington's uncle. 5. Marion is female. 6. Edd is male. 7. Bird is male.
From the information given: - Edd is Washington's father, making Washington Edd's son. - Bird is Washington's uncle, meaning Bird is either Edd's brother or Edd's brother-in-law.
Since Marion is Edd's sister, and Bird is Washington's uncle, Bird must be Edd's brother (because uncles are typically brothers of a child's parent).
Therefore, Bird is Marion's brother as well.
Marion should address Bird as her brother.
Final Answer: [brother]

Story: Friend is Ottilia's husband, Ottilia is Christine's mother, Rollie is Christine's husband. Friend is male, Ottilia is female, Christine is female, Rollie is male. How should Friend address Rollie?
Answer: To determine how Friend should address Rollie, let's break down the relationships step by step:
1. Friend is Ottilia's husband. Therefore, Friend is Ottilia's spouse. 2. Ottilia is Christine's mother. Therefore, Christine is the daughter of Ottilia and Friend. 3. Rollie is Christine's husband. Therefore, Rollie is Christine's spouse.
Now, we need to find the relationship between Friend and Rollie: - Since Christine is the daughter of Friend and Ottilia, and Rollie is Christine's husband, Rollie is the son-in-law of Friend.
Therefore, Friend should address Rollie as his [son-in-law].

Story: Young is May's husband, Abbie is May's daughter-in-law. May is female, Young is male, Abbie is female. How should Young address Abbie?
Answer: To determine how Young should address Abbie, let's break down the relationships step by step:
1. Young is May's husband. 2. May is female. 3. Young is male. 4. Abbie is May's daughter-in-law.
Since Abbie is May's daughter-in-law, this means Abbie is married to May's son. Therefore, Abbie is also Young's daughter-in-law because Young is May's husband.
So, Young should address Abbie as his daughter-in-law.
Final answer: [daughter-in-law]

Story: Leonard is Rose's father, Ella is Leonard's wife, Ella is Genevieve's mother. Rose is female, Leonard is male, Ella is female, Genevieve is female. How should Rose address Genevieve?
Answer: To determine the relationship between Rose and Genevieve, let's break down the information given in the story step by step:
1. Leonard is Rose's father. 2. Ella is Leonard's wife. 3. Ella is Genevieve's mother. 4. Rose is female. 5. Leonard is male. 6. Ella is female. 7. Genevieve is female.
From the above points, we can deduce the following relationships: - Since Leonard is Rose's father and Ella is Leonard's wife, Ella is Rose's mother. - Ella is also Genevieve's mother, which means Rose and Genevieve share the same mother. - Therefore, Rose and Genevieve are siblings.
Since both Rose and Genevieve are female, Rose should address Genevieve as her sister.
Final answer: [sister]

Story: {extracted instance path}
Answer:

Table 34: PoT-LLM prompt template for SPARTUN.

Given a story about spatial relations among objects, answer the relation between two queried objects step by step. The answer could only be one of following: [far, in, touch, has, covered_by, right, overlap, front, behind, cover, left, disconnected_from, below, above, near]. 'inside and touching' refers 'covered_by'. 'inside' and 'within' and 'inside' refers 'in'. 'contain' refers 'has'. If the sentence is describing clock-wise information, then 3 denotes right, 6 denotes below, 9 denotes left, and 12 denotes above. If the sentence is describing cardinal directions, then north denotes above, east denotes right, south denotes below, and west denotes left. There could be multiple answers. Wrap all your answers in brackets. Example: [above, behind]. A 'Reasoning chain' with respect to the question is provided to help solve the question. Please take it as the important reference. If the 'Reasoning chain' is None, then extact the reasoning chain yourself.

Story: A box called one covers a medium green apple. Covered by another box called two there is this box. Box two has a medium orange apple which touches a yellow apple. Box two covers the yellow fruit. Where is box two regarding box one?
Reasoning chain: box one is covered by box two.
Answer: To determine the spatial relation between box two and box one, we need to analyze the given reasoning chain step by step.
1. Box one is covered by another box called two.
Therefore, box two covers box one.
The final answer: [cover]

Story: A midsize orange rectangle is inside and touching a box named DDD. Above and in front of box DDD is another box named EEE. Box DDD is disconnected from and near to this box. A midsize orange rectangle is over and touches another midsize orange rectangle. Midsize orange rectangle number one is within box EEE. Box EEE covers midsize orange rectangle number two. Where is DDD relative to midsize orange rectangle number two?
Reasoning chain: box EEE is above and in front of box DDD, box EEE covers midsize orange rectangle number two.
Answer: To determine the relation between DDD and midsize orange rectangle number two, let's follow the reasoning chain step by step:
1. **Identify the position of box EEE relative to box DDD:** - The story states: "Above and in front of box DDD is another box named EEE." - This means box EEE is positioned above and in front of box DDD.
2. **Identify the position of midsize orange rectangle number two relative to box EEE:** - The story states: "Box EEE covers midsize orange rectangle number two." - This means midsize orange rectangle number two is within or under box EEE.
3. **Determine the position of DDD relative to midsize orange rectangle number two:** - Since box EEE is above and in front of box DDD, and box EEE covers midsize orange rectangle number two, it implies that midsize orange rectangle number two is above and in front of box DDD.
Therefore, the relation between DDD and midsize orange rectangle number two is that DDD is behind and below midsize orange rectangle number two.
Final answer: [behind, below]

Story: A medium triangle, a big black square and a big circle are in a block called AAA. The big black square is behind the big circle and is in front of the medium triangle. In front of and touches a small black triangle there is this thing. Block AAA has the small black triangle. This block has a small blue square. Behind the medium triangle there is the small black triangle. Behind the big circle is the medium object. The small blue square is in front of the object which was in front of the medium thing. Under the big circle is this shape. What is the position of the medium object regarding the small blue square?
Reasoning chain: medium triangle is behind the big circle, small blue square is in front of big circle.
Answer: To determine the position of the medium object regarding the small blue square, let's follow the reasoning chain step by step: 1. The medium triangle is behind the big circle. 2. The small blue square is in front of the big circle. 3. Since the medium triangle is behind the big circle and the small blue square is in front of the big circle, the medium triangle must be behind the small blue square.
Answer: [behind]

Story: {story}
Reasoning chain: {extracted instance path}
Answer:

Table 35: PoT-LLM prompt template for Chinese kinship.

You are given a question about chinese kinship relations, please answer the question step by step. ansaers include but not limited to chinese kinship titles: (从/表/堂)侄子,侄女,女婿,儿媳,岳父,岳母,妹夫,姐夫,伯公,叔公,etc. Wrap your final answer in square brackets []. If more than one relation is correct, separate the relations by comma, like: [舅表哥,舅表弟]. A 'Reasoning chain' with respect to the question is provided to help solve the question. Please take it as the important reference. If the 'Reasoning chain' is None, then extact the reasoning chain yourself.

Question: 小北最近对家谱的研究产生浓厚兴趣，在整理家族关系时，他发现自己的孙子小明有一位姑妈名叫小花，而小花的奶奶小丽是小北祖辈亲属中的一员。在一次家族聚会上，小北得知他的一个远房亲戚小顾，实际上是小丽的弟弟。根据上述信息，小北可能称呼小顾为?
Reasoning_chain: 小北的孙子是小明，小明的姑妈是小花，小花的奶奶是小丽，小丽的弟弟是小顾。其中，小欣是男的。
Answer: To determine how 小北 would address 小顾, let's break down the relationships step by step:
1. 小北 has a grandson named 小明. 2. 小明 has an aunt (姑妈) named 小花. 3. 小花's grandmother is 小丽. 4. 小丽 is a member of 小北's ancestral generation. 5. 小丽 has a brother named 小顾.
Given that 小丽 is 小花's grandmother, 小丽 must be 小北's mother or mother-in-law. Since 小丽 is 小花's paternal grandmother, 小丽 is 小北's mother.
Thus, 小顾, being 小丽's brother, would be 小北's maternal uncle (舅舅).
Therefore, 小北 would refer to 小顾 as [舅舅].

Question: 在小采的生日宴会上，家人们欢聚一堂，庆祝气氛热烈。小采是一位乐于助人的男性，他总是喜欢带领家人们一起参与各种社会活动。当天，小采的女儿小美邀请了她的嫂子小丽一同参加宴会。小丽和她的老公小帅也都到场了。在聊天中，大家提到了小伶，她是小帅的母亲。在这个喜庆的场合，小采可能称呼小伶为?
Reasoning_chain: 小采的女儿是小美，小美的嫂子是小丽，小丽的老公是小帅，小帅的母亲是小伶。其中，小采是男的，小伶是女的。
Answer: To determine how 小采 would address 小伶, let's verify step by step: 1. 小采的女儿是小美。 2. 小美的嫂子是小丽。 3. 小丽的老公是小帅。 4. 小帅的母亲是小伶。 5. 小采是男的，小伶是女的。
从以上信息可以推断： - 小采的女儿小美的嫂子小丽是小采的儿媳。 - 小丽的老公小帅是小采的儿子。 - 小帅的母亲小伶是小采的妻子。
因此，小采可能称呼小伶为[妻子]。

Question: 小孟的女儿小郁忙着与来宾们打招呼，而小孟则在一旁和她的老公小华交谈。小华提到了他的父亲小闵也即将到来。那么小郁可能称呼小闵为什么?
Reasoning_chain: 小孟的女儿是小郁，小孟的老公是小华，小华的父亲是小闵。其中，小闵是男的，小郁是女的。
Answer: To determine how 小郁 would address 小闵, let's analysis step by step: 1. 小孟的女儿是小郁。 2. 小孟的老公是小华。 3. 小华的父亲是小闵。 4. 小闵是男的，小郁是女的。
根据以上关系链，小闵是小华的父亲，因此是小郁的爷爷。
Answer: [爷爷]

Question: {story}
Reasoning_chain: {extracted reasoning path}
Answer:

Table 36: Target Ambiguities found in CLUTRR.

| Story | Answers |
|---|---|
| Ellsworth played chess with his brother Nick. Ellsworth took his son Tony to the park to feed the squirrels. Tony and his grandmother Daisie went to the science museum. They both had fun, and learned some things, too. What should Nick address Daisie? | mother-in-law, mother |
| Hampton bought to dress for his father Chester Hampton and his sister Serena went out for ice cream. Serena bought her grandfather, Orville, a tie for his birthday. Travis likes to visit his sister. Her name is Rachael. What should Chester address Orville? | father, father-in-law |
| Hessie's daughter Maymie went to grab dinner. Hessie's husband, Nicholas, was not happy about it. Maymie made a cake for her grandfather, Elizabeth. Nicholas went to lunch with his wife Hessie. What should Nicholas address Elizabeth? | father,father-in-law |

