# OpenReview forum: "Extracting and Following Paths for Robust Relational Reasoning with Large Language Models"
_TMLR — Accepted by TMLR_

### Review · Reviewer_1wRc · 2025-09-20

**Summary Of Contributions:**

The paper introduces Path-of-Thoughts (PoT), a novel neuro-symbolic framework designed to improve relational reasoning in large language models (LLMs). PoT decomposes complex reasoning tasks into three stages:\
Graph Extraction: A single LLM call extracts entities, relations, and attributes from natural language text to construct a task-agnostic graph.\
Path Identification: Relevant reasoning paths between queried entities are identified within the graph.\
Reasoning: An external reasoner (LLM or symbolic solver) infers answers based on the identified paths.\
Key Strengths:\
Novelty: First work to explicitly mitigate LLM extraction errors via multi-path reasoning.\
Effectiveness: Strong empirical results across diverse datasets and model backbones.\
Efficiency: Low computational overhead despite two-stage LLM usage.\
Generalizability: Applicable to both spatial and kinship reasoning, including challenging non-English contexts.\
Key Weaknesses:\
Dependence on Symbolic Rules: For symbolic reasoning, performance depends on manually crafted domain rules, which may not scale to very large or complex relation sets.\
Limited Error Analysis: While robustness is demonstrated, deeper analysis of failure modes (e.g., path selection errors) is lacking.

**Audience:**

Yes

**Audience Explanation:**

The paper introduces Path-of-Thoughts, which decomposes relational reasoning into graph extraction, path selection, and inference, and achieves strong results across benchmarks with robustness to extraction noise. These topics align with TMLR’s focus on LLM reasoning and neuro-symbolic methods.

**Broader Impact Concerns:**

The work presents a general framework for improving relational reasoning in LLMs and does not raise significant ethical concerns.

**Claims And Evidence:**

Yes

**Claims Explanation:**

1. Extensive experiments on four diverse datasets with varying complexity.\
2. Comparisons against strong baselines (IO, Few-Shot, CoT, CoT-SC, LLM-ASP).\
3. Use of multiple LLM backbones (GPT-3.5, GPT-4, GPT-4o, Llama3, Claude-3.7).\
4. Robustness analysis against synthetic noise and extraction errors.\
5. Detailed prompt designs and ablation on extraction performance (Table 5).\
6. Computational cost analysis showing practical efficiency.

The results are clearly presented in tables and figures, and the methodology is thoroughly described.

**Requested Changes:**

Critical for Acceptance:\
Error Analysis: Provide a qualitative analysis of failure cases (e.g., incorrect path selection, reasoning errors) to better understand limitations.\
Symbolic Rule Scalability: Discuss the limitations of symbolic reasoning in domains with very large relation sets (e.g., Chinese kinship) and propose or evaluate potential solutions (e.g., rule learning, hybrid approaches).

Would Strengthen the Work:\
Human Evaluation: Include human evaluation for extraction quality or reasoning correctness, especially for ambiguous cases.
Broader Domains: Test PoT on at least one additional domain (e.g., temporal reasoning, social networks) to demonstrate broader applicability.\
Better Task Allocation of LLMs: \
The article mentions the application of LLM in reasoning, which can refer to the article " HCTA:Hierarchical Cooperative Task Allocation in Multi-Agent Reinforcement Learning". This article introduces a novel framework called Hierarchical Cooperative Task Allocation (HCTA) that is based on long-term behavior analysis.

---

> ### Author Response · Authors · 2025-10-29
> **Responses to weaknesses and requested changes**
>
> We thank the reviewer for valuable comments and suggestions. The following are our responses with respect to each weakness and requested change.
>
> **Weaknesses**
>
> > Dependence on Symbolic Rules: For symbolic reasoning, performance depends on manually crafted domain rules, which may not scale to very large or complex relation sets.
>
> **Requested change**
>
> > Symbolic Rule Scalability.
>
> We added a limitation section (section 6) to discuss this limitation along with others.
>
> A potential solution is to apply frameworks such as [1] to build and refine symbolic rule sets with LLM for new domains with large relation sets. However, such solution is out of the scope of our paper.
>
> Consequently, we recommend PoT-Symbolic for better accuracy and efficiency in domains with a small, well-defined relation set, and suggest PoT-LLM when scalability and generality are prioritized.
>
> [1] Yang, Yang, et al. "RLIE: Rule Generation with Logistic Regression, Iterative Refinement, and Evaluation for Large Language Models.  arXiv preprint arXiv:2510.19698 (2025)"
>
> **Weaknesses**
>
> > Limited Error Analysis: While robustness is demonstrated, deeper analysis of failure modes is lacking.
>
> **Requested change**
>
> > Error Analysis: Provide a qualitative analysis of fail cases
> >
> > Human Evaluation: Include human evaluation for extraction quality or reasoning correctness, especially for ambiguous cases.
>
> We have already included the three target ambiguity cases we found during our case studies in Table 33 of the original paper. Please refer to Table 36 of the revised paper for those ambiguity cases.
>
> We have already included three types of extraction errors that lead to failure cases in Table 23 of the original paper, including incorrect relation extraction, pronoun errors, and incorrect gender. Those extraction errors, especially incorrect relation extraction, could lead to incorrect path selection between the queried entities.
>
> Moreover, we have observed incorrect reasoning errors during our PoT-LLM experiments with older LLMs, in which the LLM incorrectly infers kinship relationships. For example, in the story, we correctly identify Nicolas as Clarence's grandson and June's sister. But the LLM draws the wrong conclusion that `June is Clarence's daughter'.
>
> We included additional extraction and reasoning failures in Tables 25 and 26 of the revised paper, together with failure reasons.
>
>
> **Requested change**
>
> > Broader Domains: Test PoT on at least one additional domain (e.g., temporal reasoning, social networks) to demonstrate broader applicability.
>
> In this work, we explicitly targeted relational reasoning, where each relationship can be represented as a reasoning chain of other relationships (e.g., kinship or spatial relations). This assumption is fundamental to the Path-of-Thought (PoT) framework, as it enables reasoning through explicit relational paths. However, domains such as temporal reasoning or social networks do not always satisfy this property. Their relations are often event-dependent and dynamic, making them conceptually distinct from our targeted reasoning paradigm. Nevertheless, extending PoT to non-chain-structured reasoning domains is an interesting future direction, but it lies beyond the current scope of this work.
>
> We discussed the limitations outlined above in Section 6 of the revised paper.
>
> **Requested change**
>
> > The article mentions the application of LLM in reasoning, which can refer to the article " HCTA:Hierarchical Cooperative Task Allocation in Multi-Agent Reinforcement Learning".
>
> We thank the reviewer for providing this critical reference. It shows that decomposing a challenging task into more manageable subtasks could benefit other research domains beyond relational reasoning. We cite this paper within our paper. Please refer to the introduction section of the revised paper for more information.

---

> > ### Comment · Reviewer_1wRc · 2025-11-12
> > **This is a well-structured and thoughtful response to the reviewer's comments.**
> >
> > The authors have successfully addressed the concerns. The revisions—particularly the new Limitations section and expanded error analysis—significantly improve the paper's depth, clarity, and scholarly value. The responses demonstrate a clear understanding of the feedback and a commitment to enhancing the work. This should satisfactorily resolve the reviewer's points and strengthen the case for acceptance.

---

### Review · Reviewer_6UKo · 2025-09-30

**Summary Of Contributions:**

This paper introduces Path-of-Thoughts, a framework designed to improve the multi-hop relational reasoning capabilities of LLMs. The core idea is to decompose the reasoning process into three distinct stages. First, a graph extraction module uses a single LLM call to parse a natural language story into a graph of entities and their relationships. Second, a path identification module finds the specific chain(s) of relationships in the graph that connect the entities mentioned in the query. Finally, a reasoning module takes this isolated path and infers the final answer. The authors propose two versions of this final stage: one using an LLM and another using a symbolic solver. The method is evaluated on four relational reasoning datasets and shows significant performance improvements over various prompting baselines and a neuro-symbolic baseline.

Strengths:
- The three-stage decomposition is logical and intuitive. The idea of separating the extraction of all facts from the identification of the relevant reasoning path is a intuitive way to handle the problem.
- The robustness analysis is a strength of the paper. The experiments where noise is deliberately injected into the extracted graph provide evidence that PoT's approach of isolating the reasoning path makes the symbolic reasoner more resilient than the baseline LLM-ASP method.

Weaknesses:
- My main concern lies on the longer-term relevance of the research question studied. The paper's own results show a clear trend: the performance gap between the proposed method and simpler baselines narrows as the underlying LLM becomes more capable. Given that the experiments rely on models that are now over a year old (e.g., GPT-4o, released May 2024), it is difficult to assess whether the significant performance gains are a fundamental advantage of the PoT framework or an effective scaffold for a previous generation of models. It remains an open question whether this explicit decomposition is still required for the latest state-of-the-art LLMs, which may have overcome the specific reasoning limitations PoT aims to solve.
- The novelty feels somewhat incremental to me. The general approach of using an LLM to extract a knowledge graph which is then passed to a symbolic reasoner is a well-established pattern in neuro-symbolic AI. The key innovation here is the intermediate path identification step.
- The problem definition in Section 2 is purely descriptive and lacks formal, mathematical notation. This makes the task setup feel less rigorous and can create ambiguity. A formalized definition would provide a stronger theoretical grounding for the work.

**Audience:**

Yes

**Audience Explanation:**

The proposed approach is simple to understand and implement, and the results are strong on established benchmarks, which makes it a solid contribution to the field. Researchers on multi-hop reasoning would be interested in this paper.

**Broader Impact Concerns:**

I do not have any broader impact concerns regarding this work. The research is foundational, aimed at improving the reasoning capabilities of AI models on abstract, synthetic tasks.

**Claims And Evidence:**

No

**Claims Explanation:**

The paper's central claim, PoT surpasses baselines, is well-supported by the results presented in the tables. However, the convincingness of this evidence is tempered by the timeliness of the models used.

While the results are internally valid for the models tested, they do not provide fully convincing evidence that PoT offers an enduring advantage over simpler prompting strategies when applied to the current state-of-the-art. The claims would be substantially more compelling if the performance gap were shown to persist on more recent models.

**Requested Changes:**

The most critical change is to evaluate on State-of-the-Art LLMs. The biggest question hanging over this work is its relevance in the face of ever-improving models. To demonstrate the continued need for this framework, I believe it is critical for the authors to re-run their key experiments using one or two of the most powerful, recently released LLMs like GPT-5 and Qwen3 family. If the performance gap between PoT and simpler baselines like CoT remains significant, it would strongly validate the paper's contribution. If the gap shrinks considerably, this is an important finding that should be discussed in the paper's limitations.

The following changes are needed to position the paper's contribution and relevance to the field:
1. I strongly suggest that the authors revise Section 2 to include a more formal definition of the relational reasoning task. This would involve using notation to define the input space (stories, entities, relations), the output space, and the objective. This would improve the scientific rigor of the paper.
2. Revise the novelty claim. The claim of being the "first work to deal with LLM hallucinations and input ambiguities through path identification" feels too narrow. "Path identification" is a term defined within the paper itself. I would recommend rephrasing this to situate the contribution more clearly within the existing literature on neuro-symbolic reasoning, focusing on how isolating query-relevant subgraphs improves robustness over prior methods that might feed the entire set of extracted facts to a solver.
3. The current cost analysis is based on only 50 samples from a single, complex subset of one dataset. To make these results more generalizable, I would like to see this analysis expanded to cover at least one full test set. This is required to provide greater confidence in the efficiency claims.
4. For clarity, I suggest replacing the term "I/O prompting" with a more standard name like "Zero-shot" or "Vanilla Prompting".
5. The authors correctly note that writing the symbolic rule set for complex domains like Chinese kinship is infeasible. This is a very important practical limitation of the PoT-Symbolic approach. I think this point deserves more prominent discussion in the main body of the paper (perhaps in a dedicated limitations section) as it highlights the key trade-off between the PoT-LLM and PoT-Symbolic variants.

---

> ### Author Response · Authors · 2025-10-29
> **Responses to weaknesses and requested changes**
>
> We thank the reviewer for valuable comments and suggestions. The following are our responses with respect to each weakness and the requested change.
>
> **Weaknesses**
>
> > Experiments rely on models that are now over a year old. It remains an open question whether this explicit decomposition is still required for the latest state-of-the-art LLMs
>
> **Requested change**
>
> > evaluate on State-of-the-Art LLMs.
>
> We conducted the experiments with GPT-5 as the backbone LLM. We notice that GPT-5's reasoning effort is adjustable, which could affect the reasoning task we are trying to solve. Therefore, we try GPT-5 with two reasoning effort settings: minimal and medium. The results are shown in Table 2 of the revised paper.
>
> Although our method performs on par with baselines under medium reasoning effort, it shows improvements when the reasoning effort is minimal, requiring much fewer tokens overhead than in medium reasoning effort settings. This demonstrates that our design effectively preserves reasoning accuracy even with reduced model reasoning effort. Please refer to Table 2 and the 'Full Pipeline Performance' subsection of section 5.2 of the revised paper for more details.
>
> **Weaknesses**
>
> > The novelty feels somewhat incremental. The key innovation here is the intermediate path identification step.
>
> **Requested change**
>
> > Revise the novelty claim.
>
> We revised the claim in the introduction to more precisely reflect the novelty of our work. The new novelty claim is: 'PoT is the first work that enhances relational reasoning tasks through identifying query-relevant subgraphs before reasoning.'
>
>
> **Weaknesses**
>
> > The problem definition in Section 2 is purely descriptive and lacks formal, mathematical notation.
>
> **Requested change**
>
> > Revise Section 2 to include a more formal definition of the relational reasoning task.
>
> We revised the problem definition by adding notations to make it more formal and interpretable. Please refer to Section 2 of the revised paper for details.
>
> **Requested change**
>
> > The current cost analysis is based on only 50 samples from a single, complex subset of one dataset. To make these results more generalizable, I would like to see this analysis expanded to cover at least one complete test set.
>
> We reran the experiment with all samples from the k=3 subset of the Stepgame dataset with GPT-4o as backbone LLM. The results are presented in Tables 3 and 4 of the revised paper.
>
> Despite involving two LLM calls, PoT-LLM maintains latency comparable to prompting baselines such as IO and CoT (3.41 s vs. 2.93 s and 3.03 s, respectively). This efficiency stems from the lightweight graph extraction stage (1.36 s) and the negligible cost of path identification (< 0.001 s). The symbolic variant, PoT-Symbolic, further reduces total runtime to 2.06 s by replacing the reasoning step with explicit logical inference. Token-wise, PoT-LLM produces a modest number of output tokens across modules (56 from graph extraction and 148 from reasoning), resulting in an overall output count lower than CoT. Meanwhile, PoT-Symbolic achieves the lowest overall token generation because it requires no LLM reasoner. Overall, these results confirm that our method, PoT, introduces negligible computational overhead while enhancing the reasoning performance. Please refer to Tables 3 and 4 of the revised paper for more details.
>
>
> **Requested change**
>
> >  I suggest replacing the term "I/O prompting" with a more standard name like "Zero-shot" or "Vanilla Prompting"
>
> We thank the reviewer for this suggestion. We replaced the term `I/O prompting' with 'Zero-shot' in our Tables for better clarity.
>
> **Requested change**
>
> > writing the symbolic rule set for complex domains like Chinese kinship is infeasible. This is a significant practical limitation of the PoT-Symbolic approach, which deserves more prominent discussion in the main body of the paper, as it highlights the key trade-off between the PoT-LLM and PoT-Symbolic variants.
>
> We added a limitation section (section 6) to discuss these limitations, along with others.
>
> A potential solution is to apply frameworks such as [1] to build and refine symbolic rule sets with LLM for new domains with large relation sets. However, such a solution is out of the scope of our paper.
>
> Consequently, we recommend PoT-Symbolic for better accuracy and efficiency in domains with a small, well-defined relation set, and suggest PoT-LLM when scalability and generality are prioritized.
>
> [1] Yang, Yang, et al. "RLIE: Rule Generation with Logistic Regression, Iterative Refinement, and Evaluation for Large Language Models.  arXiv preprint arXiv:2510.19698 (2025)"

---

### Review · Reviewer_1SGm · 2025-10-14

**Summary Of Contributions:**

This paper proposes path of thought a three-stage framework for relational reasoning that decomposes reasoning tasks into 1) graph extraction from natural language 2) path identification between query entities using depth-first search, and 3) reasoning over extracted paths. The authors evaluate PoT on four relational reasoning benchmarks using multiple LLMs.

# Strengths
- The explicit path identification stage is an interesting architectural contribution to reasoning
- Comprehensive empirical evaluation across 4 datasets and multiple LLM backbones
- Thorough prompt engineering documentation in appendix
- Practical efficiency analysis (time and token usage)

# Weaknesses
- Some overclaims (see below)
- Missing discussion and critical analysis
- No ablation study to back up some claims
- Single-trial runs means no confidence intervals or error bars

**Additional Comments:**

1. Page 1 states: "Prior studies (Tolman, 1948; Whittington et al., 2022; Garvert et al., 2017) have shown that humans, unlike LLMs, can create 'cognitive maps'..." None of these papers discuss LLMs. So I'm not sure where the "unlike LLMs" comment comes from, if it is the authors' addition it should be separated out to not appear as if this is a claim from those papers.
2. Figure 3 appears within references section, this is strange placement. A simple latex formatting fix should address it.

**Audience:**

Yes

**Audience Explanation:**

The paper addresses an important problem (robust relational reasoning with LLMs) that is relevant to the TMLR community.

**Broader Impact Concerns:**

No significant concerns.

**Claims And Evidence:**

No

**Claims Explanation:**

1. The paper repeatedly claims PoT is "task-agnostic" and doesn't require domain-specific knowledge (e.g., Abstract: "task-agnostic graph extraction" Page 2: "does not rely on any predefined rules or domain-specific symbolic formalization"). However, when looking at the actual prompts (Tables 24-27, Appendix A.3) reveals that they actually have predefined (hard-coded) relations. These prompts encode substantial domain expertise and essentially the same inductive biases as symbolic rules, just expressed in natural language. I find the "task-agnostic" claim to be misleading. Note that here I am not talking about PoT vs. baseline prompts, I understand the baselines also receive similar domain knowledge, the concern here the "task agnostic" claim.
2. It is not clear if path identification improves robustness. There would be a few ways to demonstrate it, perhaps the most straightforward is a simple ablation study of PoT with and without the path identification module. In the current results, the robustness experiments compare PoT-Symbolic vs LLM-ASP, which differ in both graph structure and path identification so we cannot isolate which component drives improvements
3. I understand that cost is high so single runs are often the only option. However, they do weaken the significance of the results and some performance improvements seem marginal (possibly within noise). This might be an acceptable trade-off but there is no discussion of limitations or a discussion section at all, so the authors have not addressed this.
4. The claim that this is the "First work to deal with LLM hallucinations through path identification" requires some nuance. Multiple prior work addresses LLM errors (self-consistency, self-correction, uncertainty quantification) and uses multiple reasoning paths (CoT-SC, Tree-of-Thoughts). What seems to be novel is the explicit path extraction via DFS on relational graphs for multi-hop reasoning. The paper should do more to identify what is genuinely new vs. what is building on prior work.

**Requested Changes:**

Critical
1. Reframe "task-agnostic" claims in the paper i.e. the claims PoT doesn't require domain-specific knowledge, but prompts encode substantial domain expertise (predefined vocabularies, mapping rules, attribute schemas).
2. Add a Discussion section (or provide critical analysis somewhere in the paper). Conclusion is only 11 lines and just restates contributions. There needs to analysis of things like: when does PoT not work? What are critical assumptions? What are the limitation of the current experimental set up? Boundary conditions? Why does path identification help? Under what conditions is it beneficial vs. harmful? How to apply to new domains? What engineering effort is required? When to choose PoT-LLM vs. PoT-Symbolic?
3. Add ablation study for path identification to isolate the contribution of path identification vs. graph extraction alone

Strengthen
1. Repeat (at least some) of the results so we see variance and error bounds.
2. Report distribution of path counts across datasets, correlation between path availability and performance, analysis of path redundancy
3. Release implementation, code and non-public data

---

> ### Author Response · Authors · 2025-10-29
> **Responses to weaknesses and requested changes**
>
> We thank the reviewer for valuable comments and suggestions. The following are our responses with respect to each weakness and requested change.
>
> **Weaknesses**
>
> >   PoT being “task-agnostic” is misleading, since the prompts (Tables 24–27) contain predefined, domain-specific relations that effectively encode symbolic knowledge in natural-language form.
>
> **Requested change**
>
> > Reframe "task-agnostic" claims in the paper
>
> We have modified the "task-agnostic" claims in the abstract and introduction sections to reflect the implementation of our pipeline accurately.
>
> **Weaknesses**
>
> > It is not clear if path identification improves robustness. Currently, the robustness experiments comparing PoT-Symbolic vs LLM-ASP cannot isolate which component drives improvements
>
> **Requested change**
>
> > Add an ablation study for path identification to isolate the contribution of path identification vs. graph extraction alone
>
> We apologize for not clearly demonstrating the robustness of the path identification module.
>
> In original robustness experiments that compare PoT-Symbolic vs LLM-ASP, we didn't add seven types of noise into the text-based story. Instead, we add noise directly to the graph structure underlying the stories, as provided by the CULTRR dataset, thereby eliminating the contribution of the graph extraction module. Since both LLM-ASP and PoT-symbolic applied the same symbolic reasoner, the only difference would be the path identification module, meaning that the performance improvements shown in Figure 2 of the original paper indeed come from the path identification module.
>
> To better demonstrate the effectiveness of the path identification module, we added ablation studies over the original Stepgame and CLUTRR datasets. The results are shown in subsection 'Ablation Study on Path Identification module' of section 5.2 of the revised paper. Furthermore, we rewrote and merged the robustness experiments into this new subsection to better demonstrate robustness. It is shown that the path identification module is critical for boosting PoT-LLM performance and addressing potential noise in the input graph structure.
>
> **Weaknesses**
>
> > Single runs of experiments weaken the significance of the results
>
> **Requested change**
>
> > Repeat (at least some) of the results to see the variance and error bounds.
>
> We acknowledge that additional repetitions could further strengthen the empirical validation; however, due to computational budget constraints, extensive reruns were infeasible. Nonetheless, to rigorously assess the statistical significance of our results, we conducted a paired McNemar test and report the corresponding significance levels. Please refer to Tables 1 and 2 of the revised paper for details.
>
>
> **Weaknesses**
>
> > The claim "First work to deal with LLM hallucinations through path identification" requires some nuance.
>
> We revised the claim in the introduction to more precisely reflect the novelty of our work. The new novelty claim is: 'PoT is the first work that enhances relational reasoning tasks through identifying query-relevant subgraphs before reasoning.'

---

> ### Author Response · Authors · 2025-10-29
> **Responses to weaknesses and requested changes (Cont.)**
>
> **Requested change**
>
> > Add a Discussion section to analyze things like: when does PoT not work? What are critical assumptions? And other questions.
>
> We added a limitation section to discuss some of the questions, which can be found in section 6 of the revised paper.
>
> **When does PoT not work? What are critical assumptions? Boundary conditions?**
>
> While the proposed PoT framework shows strong performance in relational reasoning, it has several limitations and assumptions. PoT has only been tested on relational reasoning tasks, in which each relationship can be expressed as a composition or chain of other relationships. Its generality to non-relational reasoning tasks remains unexplored.
>
> **What are the limitations of the current experimental setup?**
>
> The experimental setup is constrained by dataset quality, as we found incorrect or incomplete ground-truth labels that may affect evaluation reliability.
>
> **Why does path identification help? Under what conditions is it beneficial vs. harmful?**
>
> We conducted an ablation study to investigate the effectiveness of the path identification module. The result can be found in the `Ablation Study on Path Identification module' subsection under section 5.2 of the revised paper.
>
> **How to apply to new domains? What engineering effort is required?**
>
> On the one hand, PoT-LLM frameworks can potentially be modified to handle new tasks that require extracting a question-relevant subgraph from an overall graph to answer the question. On the other hand, the PoT-Symbolic requires extra work to build a symbolic rule set before it can be applied to different domains. Potential solutions include hiring experts to create a rule set for a new domain, or using methods such as [1] to generate one automatically.
>
> [1] Yang, Yang, et al. "RLIE: Rule Generation with Logistic Regression, Iterative Refinement, and Evaluation for Large Language Models.  arXiv preprint arXiv:2510.19698 (2025)"
>
> **When to choose PoT-LLM vs. PoT-Symbolic?**
>
> We recommend PoT-Symbolic for well-defined domains with a small relation set, and suggest PoT-LLM for broader, more general applications.
>
> **Requested change**
>
> > Report distribution of path counts across datasets, correlation between path availability and performance, and analysis of path redundancy
>
> A subsection named 'Path Counts Analysis' is added in section A.2 of the appendix of the revised paper to analyze the impact of path counts on the accuracy.
>
> We computed the distribution of path counts across the CLUTRR dataset when using GPT-4o as the backbone LLM and average accuracy grouped by path count. As shown in the following table, only one path can be extracted from most of the stories (697/1048), with an average accuracy of 68.9\%. The accuracy increases as path counts grow and reaches a maximum of 84.6\% when the path count equals 3. Then the accuracy drops as the path count increases, which could be due to the increasing difficulty of the samples.
>
> To evaluate path redundancy, we conduct an experiment in which we identify only the shortest path and pass it to the downstream symbolic solver. The results are in the last row of the following table. The average accuracy drops notably across multiple path counts, reaching a maximum of 14.3\% for samples with path counts of 5 or more, showing the contribution of identifying multiple paths to final reasoning accuracy.
>
> | \# of chains                                 | 0   | 1    | 2    | 3    | 4    | 5+   |
> |----------------------------------------------|-----|------|------|------|------|------|
> | \# of samples                                | 39  | 697  | 175  | 13   | 75   | 49   |
> | Acc. of Identifying Multiple Paths (default) | 0.0 | 68.9 | 73.1 | 84.6 | 74.7 | 71.4 |
> | Acc. of Identifying Single Shortest Path     | 0.0 | 68.9 | 63.4 | 84.6 | 66.7 | 57.1 |
>
>
> **Requested change**
>
> > Release implementation, code, and non-public data
>
> We thank the reviewer for being interested in our implementation and data. We have released the prompt designs, experiment details, and public data used in our experiments in the appendix for reproducibility.
>
> However, we couldn't release the code implementation because it's under a patent review. We also couldn't release the non-public dataset (i.e., the Chinese kinship dataset) because it is a confidential internal dataset. We are sorry for the inconvenience this may cause.

---

### Decision · Action_Editor_yoVX · 2025-12-23

**Recommendation:** Accept with minor revision

**Additional Comments:**

Reviewers appreciated the overall direction of PoT and provided lots of critical feedback in their reviews, which authors incorporated in the revised manuscript, from ablations, to clarifications of the setting (kudos). Ultimately, reviewers scored the paper with "leaning reject/ leaning reject/ leaning accept", signaling a borderline score.

I went through all the discussion and saw that the two main reasons to reject are focusing on a) part of the presentation that makes claims about LLMs and their ability to form cognitive maps and b) the thin novelty of PoT, which ultimately boils down to a way to exchange computations for performance. I think a) can be easily addressed in a minor revision and b) cannot be a strong argument for rejection (see TMLR guidelines).

As such, I suggest the paper is accepted with minor revisions, and invite the authors to incorporate all the missing suggestions from reviewers, especially 1SGm who asks, among other things, to amend  “Prior studies (Tolman, 1948; Whittington et al., 2022; Garvert et al., 2017) have shown that humans, unlike LLMs, can create “cognitive maps” while navigating and experiencing their environments.”. Furthermore, they have to expand the LimitationS (please make the section name plural) section and discuss the trade-off of compute/performance in PoT.

**Audience:**

Yes

**Audience Explanation:**

The topic is definitely of interest for the TMLR community, as also noted by all the three reviewers.

**Claims And Evidence:**

Yes

**Claims Explanation:**

This paper deals with the problem of making LLMs reason in a more reliable way on relational problems. To this end, the authors introduce "Path of Thought" (PoT) as a multi-stage algorithm that decomposes reasoning tasks over relational domains into a graph extraction phase, a path identification phase and finally a reasoning phase over extracted paths. PoT is empirically testes on a number of relational reasoning benchmarks with multiple LLM baselines and its effectiveness sufficiently discussed.

Some reviewers such as 1SGm highlighted how the initial version of the paper was overclaiming some results or how it was missing in-depth discussion about the setting (eg the math foundation in Section 2 as highlighted by 6UKo) and the limitations of PoT. The authors rebutted by adding several paragraphs and ablation studies. As it is, I believe all claims are supported by enough evidence.